

# ESM-SnowMIP: Assessing models and quantifying snow-related climate feedbacks

Gerhard Krinner[1], Chris Derksen[2], Richard Essery[3], Mark Flanner[4], Stefan Hagemann[5], Martyn Clark[6], Alex Hall[7], Helmut Rott[8], Claire Brutel-Vuilmet[1], Hyungjun Kim[9], Cécile B. Ménard[3], Lawrence Mudryk[2], Chad Thackeray[7], Libo Wang[2], Gabriele Arduini[10], Gianpaolo Balsamo[10], Paul Bartlett[2], Julia Boike[11], Aaron Boone[12], Frédérique Chéruy[13], Jeanne Colin[12], Matthias Cuntz[14], Yongjiu Dai[15], Bertrand Decharme[12], Jeff Derry[16], Agnès Ducharne[17], Emanuel Dutra[18], Xing Fang[19], Charles Fierz[20], Josephine Ghattas[21], Yeugeniy Gusev[22], Vanessa Haverd[23], Anna Kontu[24], Matthieu Lafaysse[25], Rachel Law[26], Dave Lawrence[28], Weiping Li[27], Thomas Marke[29], Danny Marks[30], Olga Nasonova[22], Tomoko Nitta[9], Masahi Niwano[31], John Pomeroy[19], Mark S. Raleigh[32], Gerd Schaedler[33], Vladimir Semenov[34], Tanya Smirnova[32], Tobias Stacke[35], Ulrich Strasser[29], Sean Svenson[34], Dmitry Turkov[36], Tao Wang[37], Nander Wever[20,38], Hua Yuan[15], and Wenyan Zhou[27]

[1] CNRS, Université Grenoble Alpes, Institut de Géosciences de l'Environnement (IGE), 38000 Grenoble, France
[2] Climate Research Division, Environment and Climate Change, Toronto, Canada
[3] School of GeoSciences, University of Edinburgh, Edinburgh EH9 3FF, UK
[4] Department of Climate and Space Sciences and Engineering, University of Michigan, Ann Arbor Michigan 48109, USA
[5] Helmholtz-Zentrum Geesthacht, Germany
[6] Hydrometeorological Applications Program, Research Applications Laboratory, National Center for Atmospheric Research, Boulder, USA
[7] Department of Atmospheric and Oceanic Sciences, University of California, Los Angeles, Los Angeles, California
[8] ENVEO - Environmental Earth Observation IT GmbH, Austria
[9] Institute of Industrial Science, the University of Tokyo, Tokyo, Japan
[10] European Centre for Medium-Range Weather Forecasts (ECMWF), Reading, UK
[11] Alfred-Wegener-Institut für Polar-und Meeresforschung, Potsdam, Germany
[12] Centre National de Recherches Météorologiques, Météo-France/CNRS, Toulouse, France
[13] LMD-IPSL, Centre National de la Recherche Scientifique, Université Pierre et Marie-Curie, Ecole Normale Supérieure, Ecole Polytechnique, Paris, France
[14] INRA, Université de Lorraine, AgroParisTech, UMR Silva, 54000 Nancy, France
[15] School of Atmospheric Sciences, Sun Yat-sen University, China
[16] Center for Snow and Avalanche Studies, USA
[17] Sorbonne Universités, UMR 7619 METIS, UPMC/CNRS/EPHE, Paris, France
[18] Instituto Dom Luiz, Faculdade de Ciências, Universidade de Lisboa, Portugal
[19] Centre for Hydrology, University of Saskatchewan, Canada
[20] WSL Institute for Snow and Avalanche Research SLF, Switzerland
[21] Institut Pierre Simon Laplace (IPSL), UPMC, 75252 Paris, France
[22] Institute of Water Problems, Russian Academy of Sciences
[23] CSIRO Oceans and Atmosphere, Canberra, ACT, Australia
[24] Space and Earth Observation Centre, Finnish Meteorological Institute, 99600 Sodankylä, Finland
[25] Météo-France - CNRS, CNRM UMR3589, Centre d'Etudes de la Neige, Grenoble, France
[26] CSIRO Oceans and Atmosphere, Aspendale, Victoria, Australia
[27] National Climate Center, China Meteorological Administration
[28] National Center for Atmospheric Research, USA
[29] Department of Geography, University of Innsbruck, Austria





[30] USDA Agricultural Research Service, USA
[31] Meteorological Research Institute, Japan Meteorological Agency
[32] Cooperative Institute for Research in Environmental Sciences, National Snow and Ice Data Center, USA
[33] Institute of Meteorology and Climate Research, Karlsruhe Institute of Technology, Germany
[34] A.M. Obukhov Institute of Atmospheric Physics, Russian Academy of Sciences
[35] Max-Planck-Institut für Meteorologie, Hamburg, Germany
[36] Institute of Geography, Russian Academy of Sciences
[37] Key Laboratory of Alpine Ecology and Biodiversity, Institute of Tibetan Plateau Research, Chinese Academy of Sciences, Beijing, China
[38] Department of Atmospheric and Oceanic Sciences, University of Colorado, Boulder, Colorado, USA

*Correspondence to*: Gerhard Krinner (gerhard.krinner@cnrs.fr)

**Abstract.** This paper describes ESM-SnowMIP, an international coordinated modelling effort to evaluate current snow schemes against local and global observations in a wide variety of settings, including snow schemes that are included in Earth System Models. The project aims at identifying crucial processes and snow characteristics that need to be improved in

snow models in the context of local- and global-scale modeling. A further objective of ESM-SnowMIP is to better quantify snow-related feedbacks in the Earth system. ESM-SnowMIP is tightly linked to the Land Surface, Snow and Soil Moisture Model Intercomparison Project, which in turn is part of the 6[th] phase of the Coupled Model Intercomparison Project (CMIP6).

## 1 Introduction

Snow is a crucial cryospheric component of the climate system. It perennially covers almost entirely the large continental ice sheets, and seasonally the Earth's sea-ice and a large fraction of the Northern ice-free continental areas. Its particular role in the Earth system is linked to its physical properties, namely its high albedo and its usually low thermal conductivity. The former gives rise to the positive snow-albedo feedback (e.g., Qu and Hall 2014; Flanner et al. 2011) that amplifies global climate variations and is thought to contribute to the observed Arctic amplification of the current global warming (e.g.,

Serreze and Barry 2011; Chapin III et al. 2005; Pithan and Mauritsen 2014; Bony et al. 2006) and to the observed amplification of global warming at high altitudes (Pepin et al. 2015; Palazzi et al. 2017). Similarly, snow acts as a "fast climate switch" on shorter (weekly and seasonal) timescales, with observed strong coupling between temperature and snow cover (Betts et al. 2014) on regional scales. Thermal insulation of underlying soil in winter strongly influences the soil thermal regime and thus the thermal state of permafrost and its carbon balance (e.g., Zhang 2005; Vavrus 2007; Gouttevin et

al. 2012; Groffman et al. 2001; Park et al. 2015; Cook et al. 2008). In addition, snow influences the ecosystem carbon balance by protecting low vegetation in winter from frost damage (Sturm et al. 2001) and by conditioning springtime onset of the growing season (Pulliainen et al. 2017). Furthermore, substantial impacts of the presence of snow cover on the atmospheric circulation have been found (e.g., Cohen et al. 2012; Xu and Dirmeyer 2011; Vernekar et al. 2010); in general, the coupling between snow and atmosphere is strongest during the melting season, and the effect of snow was found to last





well into the snow-free season because of its delayed effect on soil humidity (Xu and Dirmeyer 2011). Linked to its effect on soil humidity, snow has an obvious and profound impact on water availability in snow-dominated regions (Barnett et al. 2005), and large potential economic impacts of snowpack decrease in a warming climate can be expected regionally (e.g., Fyfe et al. 2017).

Observed global and (predominantly Northern) hemispheric trends of snow cover extent and duration over the recent decades are consistently negative and linked to the observed warming trends (e.g., Derksen and Brown 2012; Brown and Mote 2009; Déry and Brown 2007). Observed snow mass follows similar trends (Cohen et al. 2012; Räisänen 2008), except in high Northern latitudes, where warming leads to higher moisture availability (Räisänen 2008; Mote et al. 2018). Global and hemispheric seasonal snow mass, extent and snow cover duration is consistently projected to decrease with on-going

warming (e.g., Brutel-Vuilmet et al. 2013; Thackeray et al. 2016; Mudryk et al. 2017; Collins et al. 2013).

The snow modules included in the global coupled climate models which are used for producing these projections come in varying degrees of complexity, from very simple slab models with prescribed physical properties of snow to more sophisticated multi-layer models that represent processes such as snowpack compaction, liquid water percolation, snow interception and unloading by vegetation etc. in more or less detail. For example, widely varying treatments of vegetation

masking of snow are suspected to be a major reason for large intermodel variations of the intensity of the snow-albedo feedback (Qu and Hall 2014; Thackeray et al. 2018). Besides snow modules included in the land-surface parameterization packages of large coupled Earth System Models (ESMs), there is a large number of other snow models for a range of applications, their degree of complexity depending on the intended applications (e.g., Magnusson et al. 2015).

It is clear that some important physical processes affecting snow, particularly in very cold conditions, are not captured even

by the most detailed physically-based snow models (Domine et al. 2016). In addition to these physical processes, accurate representation of vegetation distribution and parameters are also found to be critical for realistic simulation of surface albedo for snow-covered forests (Essery 2013; Loranty et al. 2014; Wang et al. 2016). Interestingly, however, the behaviour of many snow models can be emulated with multi-physics models (Essery 2015; Lafaysse et al. 2017). This suggests that in spite of their large variety, "many of them draw on a small number of process parameterizations combined in different

configurations and using different parameter values" (Essery et al. 2013). This gives reason to hope that some persistent problems of snow models, and some snow-related problems in ESMs, could in fact be tackled fairly easily, sometimes simply by careful parameter choices. In ESMs, these problems include, for example, the representation of snow masking by vegetation (Essery 2013; Thackeray et al. 2015), the thermal (Cook et al. 2008; Gouttevin et al. 2012) and radiative (Thackeray et al. 2015; Flanner et al. 2011) properties of the snowpack, and a resulting persistent large uncertainty

concerning the emergent strength of the planetary snow albedo feedback (Qu and Hall 2014; Flanner et al. 2011). However, it is also clear that even if there is a large room of improvement in the snow schemes currently used in ESMs, the current knowledge of snow physics also maintains an irreducible uncertainty in snow modeling, even in the most detailed snowpack models currently available (Lafaysse et al. 2017).



Several model intercomparison exercises focusing on snow, or at least regions that are heavily influenced by the seasonal presence of snow, have been carried out in the past, notably the Programme for Intercomparison of Land-Surface Parameterization Schemes (Pilps) Phase 2d (Slater et al. 2001) and 2e (Bowling et al. 2003) and SnowMIP Phase 1 (Etchevers et al. 2004) and Phase 2 (Essery et al. 2009; Rutter et al. 2009). These intercomparisons have highlighted some

common problems of snow models such as vegetation masking and internal processes affecting snow physical properties particularly during the melting season, leading to potentially large errors in the simulated date of disappearance of the seasonal snow cover. These previous intercomparison exercises have been carried out at small scales, and one important conclusion of the most recent one of these, SnowMIP Phase 2, was that the challenge of evaluating snow models at larger scales, on which they are often applied, needed to be tackled (Essery et al. 2009).

The purpose of this paper is to present a new, already ongoing initiative aiming at evaluating a large range of snow models both at local and large scales, including in particular, but not exclusively, land surface models that are part of the ESMs contributing to the current 6th phase of the Coupled Model Intercomparison Project (CMIP6: Eyring et al. 2016), and specifically to the Land Surface, Snow and Soil Moisture Model Intercomparison Project (LS3MIP: van den Hurk et al. 2016), which is part of CMIP6. The initiative presented here, called ESM-SnowMIP, is an extension of LS3MIP including

site-scale evaluation and process studies as well as complementary, snow-specific large-scale (global) simulations and analyses.

The overall objectives and rationale of ESM-SnowMIP are presented in the following section. Section 3 describes the planned experiments and analysis strategy, presenting some initial results from the site-scale reference simulations. The discussion in Section 4 concerns the expected outcome and impacts of ESM-SnowMIP as well as possible future extensions.

**2 Objectives and rationale**

The first objective of ESM-SnowMIP is to assess the current state of the art of snow models on spatial scales ranging from the site scale to global scales. On the site scale, the availability of longer-term high-quality observations at a larger range of sites than in previous intercomparison exercises provides the opportunity for a more comprehensive assessment of the current modelling capacity in different climate settings (see the section on reference site simulations). Similarly, on the

global scale, a wealth of new large-scale observational data sets based on advanced remote-sensing techniques allows for more meaningful evaluations than has been possible in the past, as will be described in the relevant section below.

In this respect, one particular motivation of ESM-SnowMIP is to profit from the multi-model CMIP6 setting, and, at the same time, make particular snow modelling and observational expertise available to climate modelling groups that in the past have not focused their attention on the representation of snow in their coupled models. CMIP6 provides the opportunity to

evaluate the representation of the historical evolution of seasonal snow in a number of mutually consistent global simulations with varying degrees of freedom, ranging from global coupled ocean-atmosphere simulations to AMIP-type climate simulations with prescribed oceanic boundary conditions (Gates 1992) to land-surface only simulations (LMIP) forced by





observationally-based meteorological data (van den Hurk et al. 2016). Combining the evaluation of these global-scale simulations with the detailed process-based assessment at the site scale provides an opportunity for substantial progress in the representation of snow, particularly in those Earth System Models that have not been evaluated in detail with respect to their snow parameterizations. We aim at identifying the optimum degree of complexity required and sufficient in global

models to simulate snow-related processes satisfyingly on large scales, at identifying previously unrecognized weaknesses in these models and at identifying feasible ways to correct these by including relevant processes and setting model parameters judiciously. Besides the site simulations using the reference model setup, additional simulations at the same scale are planned to identify the role of specific processes or snow properties such as snow albedo and thermal conductivity.

A second objective of ESM-SnowMIP is to allow for a better quantification of snow-related global climate feedbacks. In

LS3MIP (van den Hurk et al. 2016), simulations extending the Glace-CMIP approach (Seneviratne et al. 2013) are planned to quantify the combined land-surface feedbacks involving snow and soil moisture on interannual time scales and in the context of projected future climate change. Complementary coordinated simulations in ESM-SnowMIP, described in the relevant section of this paper, aim at isolating the effect of snow from that of soil moisture. Diagnoses of snow shortwave radiative forcing as simulated by the participating ESMs, a metric of the radiative effect of snow cover within the climate

system (Flanner et al. 2011), complete this aspect of ESM-SnowMIP.

ESM-SnowMIP is part of the World Climate Research Programme (WCRP) Grand Challenge "Melting Ice and Global Consequences"[1]. As such, it is intended to ensure rapid progress in the understanding of snow-related processes and feedbacks in the global climate, and their depiction in global climate models in the context of the on-going global changes, which are characterized by a rapid decrease of the extent and mass of the global cryosphere.

**3 Experimental design and analysis strategy**

As in CMIP6, experiments in ESM-SnowMIP are tiered. Tier 1 simulations are supposed to be carried out by all participating groups provided their model structure is adapted to the specific experiment; for example it is obviously impossible for a snow model that is not part of an ESM to participate in the coupled experiments, even those that are labeled as tier 1. The number of groups participating in tier 2 simulations will necessarily be lower or equal to those participating in

tier 1 experiments; we anticipate that not all proposed tier 2 experiments will necessarily attract a sufficient number of participating groups for a meaningful multi-model analysis to be possible. The experiments proposed in ESM-SnowMIP, with links to relevant LS3MIP reference simulations where appropriate, are listed in Table 1 and described in detail in this section. We start with simulations at the plot scale, part of which have already been carried out and are currently being analysed, and then describe global distributed simulations that will be carried out by a subset of the models participating in

ESM-SnowMIP, namely global land surface models (LSM) that are also components of an ESM. Finally, we describe planned ESM experiments.

---

[1] http://www.climate-cryosphere.org/activities/grand-challenges





### 3.1 Local scale

### 3.1.1 Overview of the sites, models, forcing and evaluation data

Global snow simulations are subject to uncertainty in the meteorological data used to drive models (whether provided by

bias corrected reanalyses as in LS3MIP offline land model experiments (van den Hurk et al. 2016) or by coupling with atmospheric models as in CMIP6), global products providing vegetation and soil characteristics for model parameters are often contradictory, and global observations of snow properties for evaluation of models (e.g., for snow density and thermal conductivity) are limited. To gain more insight into the behaviour of models in coupled and uncoupled global snow simulations, ESM-SnowMIP includes experiments using high-quality driving and evaluation data from well-instrumented

reference sites. These process-based studies have been enabled by the considerable efforts of organizations maintaining the sites to compile, quality control, gap fill and publish their data. Even these reference sites cannot provide all of the input data required by the most sophisticated snow physics models, such as shortwave radiation partitioned into direct and diffuse components and aerosol deposition fluxes. Details of sites used in a first round of ESM-SnowMIP reference site simulations that have already been completed are given in Table 2, and temperature and snowfall statistics are shown in Figure 1 (the

forcing data provide separate rain- and snowfall rates). Alpine, arctic, boreal and maritime sites have been included in the first round of simulations, and a second round will introduce tundra and glacier sites. The challenges of maintaining unattended hydrometeorological measurements in cold and snowy environments and a requirement for multiple years of data limit the number of possible sites, but the range of sites and the numbers of years simulated in ESM-SnowMIP far exceed those in similar experiments for SnowMIP and PILPS2d.

Only climate and Earth System Models will be able to perform the global coupled simulations required for CMIP6, but the local uncoupled reference site simulations for ESM-SnowMIP can be performed by a wider range of models at much lower computational expense. Models that have already completed the first round of reference site simulations, listed in Table 3, include land surface schemes (LSS) of CMIP6 models, sophisticated snow physics models, hydrological models and multi-physics ensemble models.

### 3.1.2 Tier 1: Reference site simulations (Ref-Site)

Snow water equivalent (SWE) and depth measurements (and therefore also bulk snow density) are available for all of the reference sites, and several sites also have albedo, surface temperature and soil temperature measurements. As examples, Figure 2 and Figure 3 show measurements and simulations at Col de Porte, which has mild and wet winters, and Sodankylä, which is cold and dry. Observations of SWE, snow depth, surface albedo and soil temperature are within the model spread

and close to means of the model ensemble. At the warmer site, the simulations of SWE and depth spread out rapidly as snow accumulates, but most of the soil temperature simulations remain within a relatively narrow range. Some models have rather





low albedos, leading to early melt, and other models melt the snow too late. Apart from a few outliers, SWE simulations at the colder site remain tightly bunched until the spring, but there is a wide spread in winter soil temperature simulations. Some models maintain soil temperatures under snow close to 0°C, whereas other models are much too cold. For both sites, there is a strong reduction in model temperature spread as soils cool in autumn.

With several observed variables available for comparison with model outputs and several metrics that can be used for measuring the match between models and observations, there are many ways in which the reference site simulations could be evaluated and ranked. Figure 4 shows one example, in which root mean squared errors in simulated SWE have been calculated for each model at each site and normalized by the standard deviation of measured SWE for comparison between sites. A value greater than 1 for this metric shows that a model fits variations in the observations no better than the average

of all observations. Ranking models according to their average error for all sites shows that a couple of models perform well and a couple perform poorly at all sites, but most models perform well at some sites and poorly at others. Many models have larger errors for the forested sites than the open sites and larger errors for warmer sites than colder sites. The ensemble mean of the models has lower errors than the majority of the individual models at most of the sites.

### 3.1.3 Tier 2 site simulations

Snow mass balance is influenced by radiated, advected and conducted heat fluxes in the energy balance, and soil temperature is influenced by snow depth and thermal conductivity. To investigate how these influences differ between models, additional experiments are proposed with prescribed snow albedo, with prescribed aerodynamic parameters and with the thermal insulation of snow removed. These experiments have not started yet, but pilot studies have been conducted using version 2 of the Factorial Snow Model (FSM, Essery 2015); this is a multi-physics model designed to run ensembles of simulations

producing a range of model behaviours by using alternative parametrizations for snow albedo, thermal conductivity, compaction, liquid water storage and coupling with the atmosphere. The pilot studies using FSM allow validating the intended experiment setup, provide a benchmark for model spread, and facilitate the interpretation of the results.

**Fixed snow albedo (FA-Site).** Seasonal and subseasonal variations of snow albedo are substantial and strongly influence the energy balance of the snowpack. This is particularly important during the melting season when complex processes within the

snowpack lead to strong and rapid variations of albedo. Positive albedo feedbacks strongly influence melt timing. Snowmelt timing is a critical climatic variable that is often incorrectly represented in climate and dedicated snow models, but it is difficult to untangle the effects of the simulation of snow albedo from other processes because of the strong feedbacks involved. An experiment in which snow albedo is fixed to 0.7 (which approximates the CMIP5 multi-model mean peak snow albedo for non-boreal snow) will enable evaluation of the effect of seasonal snow albedo variations and biases.

Differences between an ensemble of fixed snow albedo simulations with FSM and reference simulations for Col de Porte are shown in Figure 5. Ensemble members differ widely in their responses to the removal of snow albedo feedbacks. Fixing the snow albedo prevents it from decreasing as the snow melts and delays the snowmelt. Extending the duration of snow cover





as a result delays warming of the soil in spring, leading to large temperature differences when snow remains in a fixed albedo simulation but has disappeared in the corresponding reference simulation.

**No suppression of fluxes in stable surface layers over snow (NS-Site).** Models generally calculate turbulent heat fluxes in the surface layer using exchange coefficients that only depend on surface roughness and wind speed in neutral conditions but

are reduced in stable conditions by a factor depending on a Richardson number or an Obukhov length. The strength of this decoupling between the atmosphere and the surface is a major source of uncertainty in climate models and will influence how strongly snowmelt responds to warming of the atmosphere. The coupling strength in a model could be quantified by an additional experiment in which exchange coefficients are kept fixed at neutral values, or fixed positive Richardson numbers are imposed.

FSM only has a single option for stability adjustment of the surface exchange coefficient, but ensemble members still respond differently to switching this option off, as seen in Figure 6. Snow-free soil temperatures would also be influenced, so the fix is only applied when snow is on the ground. With heat transfers predominantly being downwards from the atmosphere to snow, fixing the exchange coefficient increases the heat transfer and warms the snow surface, often leading to decreases in snow albedo and earlier melt. The soil warms rapidly in spring when the snow melts, but winter soil

temperatures can also be increased relative to the reference simulation despite the decrease in insulating snow depth because of the increased heat flux from the atmosphere.

**No thermal insulation by snow (NI-Site).** The low thermal conductivity of snow has major climatic effects on the temperature of underlying soils and heat fluxes to the atmosphere that are highly variable and often not well represented in climate models (Koven et al. 2013). This insulating effect might be quantified by an experiment in which snow is attributed a

very high (effectively infinite) thermal conductivity while its other properties (albedo, latent heat of melting, etc.) are kept unchanged. In practice, the numerical scheme of a model might become unstable for high thermal conductivities and another solution might be envisaged, such as resetting the temperature or the net heat flux at the soil-snow interface to that calculated at the snow surface.

In a pilot study, FSM simulations were found to be numerically stable with a fixed 50 W m$^{-1}$ K$^{-1}$ thermal conductivity for

snow, which is much higher than a typical range of 0.05 to 0.5 W m$^{-1}$ K$^{-1}$ (Sturm et al. 1997). Results are shown in Figure 7. Without the insulating effect of snow, the soil freezes even in the relatively mild winters at Col de Porte. Building up a cold reservoir in the soil over winter has a secondary effect of delaying snowmelt in spring. Even without the insulating effect, the high albedo of snow and energy required for snowmelt reduce the amount of energy to warm the soil, leading to a second trough in soil temperature differences between high thermal conductivity and reference simulations.

**Downscaled large-scale forcing (LSF-downscaled-Site).** Most of the mid-latitude ESM-SnowMIP reference sites were established for snow research in mountainous regions and are at higher elevations than much of the surrounding terrain. Meteorological variables in large-scale forcing datasets, such as the GSWP3 forcing provided at 0.5° spatial resolution for LS3MIP, would therefore be expected to be biased relative to in situ measurements at the sites even if they were perfect on the grid scale. Col de Porte, for example, is located at an elevation of 1325 m in the French Chartreuse Mountains but lies



within a 0.5° grid cell with an 870 m average elevation. Figure 8a shows that an FSM simulation for winter 2009-2010 at Col de Porte with GSWP3 driving data gives almost no snow accumulation; this is because temperature is overestimated, total precipitation is underestimated and snowfall is severely underestimated. Site and grid elevations for Sodankylä, in contrast, only differ by 40 m. The large-scale simulation shown in Figure 8b is not so strongly influenced by driving data biases.

Downscaling is commonly required when using regional climate predictions in hydrological impact studies. An ESM-SnowMIP experiment with bias-corrected large-scale driving data will be helpful in using reference site observations and simulations to evaluate the performance of models in large-scale simulations. Simply removing average biases in the GSWP3 driving data, with no attempt to adjust variations on interannual and shorter timescales, greatly improves the SWE simulation for Col de Porte, replacing a massive underestimate with a slight overestimate (Figure 8a). For Sodankylä,

removing the smaller long-term driving data biases has very little effect on the SWE simulation (Figure 8b).

We expect the Tier 2 site simulations with the individual models to essentially align with the FSM results presented here. However, we also expect a varying degree of sensitivity of the various models to the different model parameter and setup changes, which will allow identifying specific priorities for continuous development and improvement for each of the participating models.

**3.2 Global scale**

**3.2.1 Large-scale observational data**

Observation-based estimates of SWE and SCF are required for evaluation of historical mean and model spread from ESM-, AMIP-, and LMIP- type simulations, as well as prescribed historical SWE and SCF as required for specific experiments. For this purpose, we have developed a blended data set of snow analyses from ERA-Interim/Land (Balsamo et al. 2015),

MERRA (Rienecker et al. 2011), MERRA-2 (Reichle et al. 2017), ERA-Crocus (Brun et al. 2013), ESA-GlobSnow (Takala et al. 2011), ERA-Brown (Brown et al. 2003) and GLDAS-2 (Rodell et al. 2004). These SWE data sets were assessed previously for temporal and spatial consistency (2015), who proposed a climatology derived from a combination of these five datasets.

The rationale for using a blended suite of snow analyses is threefold. First, it provides a measure of historical observational

uncertainty given by the range of estimates from individual analyses. This is demonstrated in Figure 9, which shows the daily median and spread (5th-95th percentile) among all seven snow analyses listed above for the 1981-2010 seasonal SWE climatology. For a given day, statistics are calculated from the pooled distribution of data across the thirty-year period and across all seven data sets. As analyzed in Mudryk et al. (2015), the range across the seven snow analyses likely results from differences in the snow schemes within the land surface models, differences in precipitation and temperature in the forcing

meteorology (from various reanalyses), and the impact of satellite and weather station measurements (used in GlobSnow and ERA-Brown). Still, the illustrated spread is a useful proxy for observation-based uncertainty (which cannot be determined



when a single product is applied for evaluation) and may be used to evaluate the corresponding level of agreement from LS3MIP and ESM-SnowMIP simulations over a similar historical period.

A second reason to use a suite of analyses for MIP evaluation is that uncertainty in the fully characterized bias and error of individual data sets provides minimal reason to favor one analysis over another. Furthermore, it has been demonstrated that combinations of products have both lower bias and RMSE than individual products when evaluated over domains with in situ data (Schwaizer et al. 2016). For this reason, a blended combination of snow analyses will be used for time varying prescribed SWE simulations (see Section 3.2.2), and could also be used for evaluation of SWE output from simulations which are observationally constrained by non-snow related variables.

Finally, the use of SWE analyses also allows the definition of a mutually consistent set of SCE data by choosing a seasonal and climatological SWE threshold above which a grid cell is considered snow-covered. The spread of total NH SCE estimated from a range of thresholds between 0-10 mm is large (Figure 10; light shading) with much of the uncertainty related to very low SWE thresholds (<2 mm), for which reanalysis-based SWE persists on the land surface for physically unrealistic amounts of time. Optimization based on satellite-based observations of climatological SCE has identified 5 mm as a reasonable choice of threshold (Figure 10; dark shading) for deriving SCE from SWE.

### 3.2.2 Global land-only simulations

The global land-only simulations planned in ESM-SnowMIP are designed as complements to a reference historical land simulation (Land-Hist) currently carried out in the framework of the LS3MIP project (van den Hurk et al. 2016). The aim of this 1850-2014 simulation, using GSWP3 meteorological forcing (Kim et al. 2018), is to provide a land-only simulation carried out with the land surface modules used in the CMIP6 ESMs at the same resolution as used in the coupled model, allowing to evaluate separately the land surface components of these models and potentially attributing sources of coupled model biases to the individual coupled model components. The global land simulations planned in ESM-SnowMIP share the model setup with this Land-Hist simulation to optimize complementarity.

**Tier 1: Prescribed observed snow water equivalent (SWE-LSM).** The relationship between grid-scale snow water equivalent (SWE), fractional snow cover and hence surface albedo is complicated and very diverse solutions are presently implemented in coupled climate models. Here we propose a prescribed SWE experiment to identify LSM biases that are linked to the parameterization of surface albedo as a function of snow cover fraction (which in turn is usually a function of SWE). The aim is to evaluate the simulated grid-scale albedo in these simulations against satellite-based observations of surface albedo.

Simulated grid-scale surface albedo in the presence of snow can depend explicitly on subgrid-scale topography, parameterized patchiness, vegetation cover, snow albedo, and other factors. The vegetation cover dependence includes explicitly simulated masking of vegetation by snow or vice versa. In particular, the albedo effect of transient snow load on trees after snowfall with subsequent unloading due to wind and melting, which is sometimes represented in current-generation ESM snow modules, should not be offset by too simple a prescription of observed SWE. It should therefore be





left up to the modeling groups to decide exactly how SWE is prescribed in their models. However, the model SWE should satisfy the condition that the weekly average SWE in the model is close to the observed value (by less than 10% or so). This can, for example, be obtained by a Newtonian relaxation of SWE to the weekly average with a time constant of a few days. Other state variables of the snow module (e.g., snow internal temperature, water content, snow grain size, etc.) will have to

be adapted accordingly; again, given the diversity of snow modules, it is impossible to define here exactly how this needs to be done in general. Note that these considerations also apply for the land-only simulations of LS3MIP in which soil wetness and SWE are to be prescribed (van den Hurk et al. 2016). In cases of snow modules where an unequivocal relationship ties surface albedo to SWE, it might be sufficient to run only the albedo scheme with prescribed SWE as input.

While a number of snow analyses are available to serve as prescribed SWE, we recommend the Mudryk et al. (2015)

combined climatology (see section 0) for ESM-SnowMIP simulations. Because biases in individual products are compensated through averaging, this dataset represents an improved reference for model evaluation compared to any single component dataset (Sospedra-Alfonso et al. 2016), just like a climate model ensemble mean is often preferred over a single member.

The simulated surface albedo will be compared to surface albedo as derived from satellite observations (MODIS (Schaaf et

al. 2002), APP-x (Wang and Key 2005), GlobAlbedo (Lewis et al. 2012)). In particular, the change in the quality of the simulated surface albedo, compared to the "free" LS3MIP Land-Hist simulation and the historical CMIP6 simulation will be evaluated in order to infer the part of surface albedo errors linked to erroneous snow mass balance.

**Tier 2: Fixed snow albedo (FA-LSM).** This is a spatially distributed version of the FA-Site simulation described above. It consists of prescribing snow albedo to a fixed value of 0.7. The aim of the experiment is, similar to that of the FA-Site

simulation, to enable evaluation of the effect of seasonal snow albedo variations and biases in LSMs, although the model response will depend very much on how snow masking by vegetation is parameterized.

Simulated snow water equivalent (SWE), fractional snow cover, vegetation masking, etc. will still influence the grid-point average surface albedo. The simulation period is 1980-2014, as in the Land-Hist and SWE-LSM simulations. If possible, the fixed snow albedo value should also be used over the ice sheets. Correct prescription of snow albedo can be verified by

checking grid-scale average surface albedo in areas with deep snow cover and low vegetation. In addition, the effect of vegetation masking on surface albedo in snow-covered areas will be isolated, since the snow-vegetation parameterizations will vary between models, but snow albedo will remain fixed.

This simulation is tightly linked to the LS3MIP Land-Hist offline reference simulation. In synergy with the site simulation with prescribed snow albedo (FA-Site), comparison with the same period in the reference simulation allows evaluation of the

effect of snow albedo in terms of timing of snow melt, winter season surface temperature, energy flux partitioning and potentially as a source of model biases. In addition, the effect of vegetation masking on surface albedo in snow-covered areas will be isolated, since the snow-vegetation parameterizations will vary between models, but snow albedo will remain fixed. Again, as in FA-Site, a basic metric to evaluate the effect of prescribed snow albedo will be the duration of snow cover (in particular melt onset) in this experiment compared to the reference simulation and observation. Required



observations therefore concern snow cover seasonality, in particular snow melt dates, and general climate variables such as surface air temperature etc.

**Tier 2: No thermal insulation by snow (NI-LSM).** We propose a global LSM simulation with "infinite" snow conductivity (that is, no effective thermal insulation by snow) as a global extension of the site-scale experiment NI-Site. Again, this simulation will have an identical setup as the relevant reference simulation Land-Hist, except obviously for the prescribed snow thermal conductivity. However, the models might need to be spun up for a substantial number of years in this setup in order to achieve thermal equilibrium at the lowest soil levels.

In this global setting, simulated potential permafrost extent (that is, the permafrost extent in equilibrium with the prescribed climate and model setup; often also termed "near-surface permafrost", e.g. Lawrence et al. (2008)) and active layer thickness will be diagnosed from the thermal state of the lowermost soil layer in the simulations. It will be compared to the corresponding output of the Land-Hist reference simulation and GTN-P observations (Biskaborn et al. 2015). Required reference data are soil temperature measurements and observations and analyses of surface energy fluxes at all seasons in areas with seasonal snow cover. Again, in the multi-model context, we expect a relationship between the sensitivity of the simulated potential permafrost extent to thermal insulation by soil and diagnosed errors of the simulated near-surface permafrost extent, which we hope will be useful to identify ways for model improvement.

**Tier 2: Fixed land cover (FLC-LSM).** Previous studies show that inaccurate representation of vegetation distribution and parameters in LSMs may result in large biases in simulated surface albedo for snow-covered forests (Essery 2013; Wang et al. 2016). Most current LSMs represent vegetation from a set of Plant Functional Types (PFTs), which are usually derived from global land cover datasets (Bonan et al. 2002; Poulter et al. 2015). There are large differences among PFTs used in LSMs, which may result from the differences in the land cover datasets, the cross-walking tables used to map land cover datasets into PFTs represented in LSMs, or uncertainties in dynamic PFT simulations (Hartley et al. 2017; Poulter et al. 2011). In order to separate biases due to differences in vegetation distribution from those due to physical processes in LSMs, we propose an experiment in which models derive their PFTs from the same land cover dataset and using the same cross-walking table.

Several global land cover datasets are available with spatial resolutions ranging from 300m to 1km (Bontemps et al. 2012). The newly released European Space Agency (ESA) Climate Change Initiative (CCI) land cover datasets are developed specifically to address the needs of the climate modelling community (ESA 2017). The CCI maps include 22 level 1 classes and 15 level 2 sub-classes based on the United Nations Land Cover Classification System, which was identified as a suitable thematic legend and compatible with the PFT concept of most LSMs (Bontemps et al. 2012). While most previous land cover datasets are for a single year, the CCI datasets are available from 1992-2015 at 300m resolution (ESA 2017). The finer spatial resolution of 300m (versus 1km) makes it inherently superior for land cover mapping in heterogeneous landscapes where different datasets tend to disagree (Herold et al. 2008; Fritz et al. 2011). In addition, a cross-walking table to convert the categorical land cover classes to the fractional area of PFTs was provided with the CCI datasets (Poulter et al. 2015). We thus suggest the use of the CCI 2000 as the common land cover dataset from which to derive PFTs. Different models usually





have their own unique set of PFTs. For cases in which both the phenology type and the associated climate zone are considered, the Koeppen-Geiger climate classification can be used as in Poulter et al. (2015).

The simulation period of 1980-2014 matches the LS3MIP Land-Hist offline reference simulations. A comparison of the surface albedo with the reference simulation will isolate the impact of PFT distribution on surface albedo and associated

feedbacks in snow-covered areas. Since the PFTs are from the same land cover data in the participating models, the differences in surface albedo among the models will reveal differences in snow-vegetation interactions and other vegetation related parameterizations (e.g. leaf area index) used in the models.

### 3.2.3 Coupled global simulation: SnowMIP-rmLC

ESM-SnowMIP proposes one coupled Tier 1 experiment, which serves the purpose of quantifying snow-related feedbacks in

the global climate system on interannual time scales. It is designed to separate the effects of snow from the combined effects of snow and soil humidity, the combined effect being addressed by the LS3MIP Tier 1 coupled experiment LFMIP-rmLC (van den Hurk et al. 2016). This LS3MIP experiment uses 30-year running mean land conditions (snow and soil humidity) as simulated in a reference transient climate change experiment, and prescribes these in the LFMIP-rmLC experiment. In these runs, snow and soil moisture feedbacks on decadal and shorter timescales are muted. Comparing the LFMIP-rmLC

simulation to the appropriate scenario simulation used for prescribing the land surface conditions allows identifying these feedbacks. In the context of a transient run, additional diagnoses of geographic shifts of land-atmosphere coupling hotspots (Seneviratne et al. 2006) and changes in potential predictability related to land surface (Dirmeyer et al. 2013) can be carried out. In order to isolate the effects of snow-atmosphere coupling, we suggest carrying out a simulation in which only the snow state is prescribed from the coupled model's CMIP6 climatology (not the observed climatology).

For the SnowMIP-rmLC experiment, the LFMIP-rmLC experiment setup is modified such that only the climatological snow variables (in particular snow water equivalent) are prescribed. Soil moisture and other land surface prognostic state variables are allowed to evolve freely. Because of internal variability in the climate system, a 5-member ensemble simulation would be ideal, but this is expensive. Similar to the LFMIP-rmLC setup, we propose the first ensemble member as Tier1, and suggest 4 other ensemble members as Tier2 (see Table 1). The simulation period is the same as in LFMIP-rmLC, i.e. 1980-

2100. Correct prescription of prescribed snow can be verified easily by comparing the simulated SWE for an individual year with the simulated climatological (1980-2014) SWE of the free scenario simulation. It should be very close.

This simulation is linked to the CMIP6 historical simulation and to the LFMIP-rmLC experiments of LS3MIP. The SnowMIP-rmLC experiments will allow evaluation of the effect of snow feedbacks on interannual to decadal time scales as well as on the centennial climate change signal (since even by the end of the 21st century, the 1980-2014 average snow

conditions will be used).



The simulation will be analyzed in parallel to the LFMIP-rmLC simulations, following very closely the methodologies of Seneviratne et al. (2013). Required observations are snow cover seasonality, in particular snow melt dates, and general climate variables such as surface air temperature, circulation patterns etc.

### 3.2.4 Snow Shortwave Radiative Effect diagnosis

Another useful measure of the impact of snow on climate is the snow shortwave radiative effect (SSRE) (e.g., Flanner et al. 2011; Perket et al. 2014; Singh et al. 2015). For the purposes outlined here, SSRE is the instantaneous change in surface absorbed solar energy flux caused by the presence of terrestrial snow. The diagnosis of SSRE provides a precise, overarching measure of the snow-induced perturbation to solar absorption in each model, integrating over the variable influences of vegetation masking, snow grain size, snow cover fraction, soot content, and other factors. SSRE is also a useful measure for

climate feedback analysis, and has a direct analog in the widely used "cloud radiative effect". To enable us to calculate and to analyze inter-model differences of SSRE and their causes, participating modelling groups are requested to provide specific gridded output (see below) from their LS3MIP Land-Hist and Land-Future simulations, as well as from the ESM- SnowMIP FA-LSM and SWE-LSM simulations. Ideally, these output fields should also be provided for one or more of the coupled atmosphere-ocean simulations, preferably from the historical reference run.

SSRE can be calculated in a land surface model through the following procedures:

1. Conducting an additional surface albedo calculation at each model timestep with zero snow. This implies setting to zero the mass of snow on ground, mass of snow in vegetation canopy, and snow cover fraction, but only for the purpose of this diagnostic albedo calculation. It should have no effect on the prognostic snow simulation.

2. Calculating net and reflected surface solar energy fluxes, each model timestep, using the diagnostic albedo from (1) and using the same surface downwelling (incident) flux that would otherwise be used to calculate solar heating.

3. Archiving the diagnostic calculations from (1) and (2) at the same frequency as other model output (e.g., daily or monthly).

The following gridded fields should be provided from the model:

- Net surface shortwave irradiance calculated without snow (rss_nosno)
- Mean shortwave surface albedo calculated without snow (albs_nosno)

Net surface solar energy flux in the absence of snow can then be differenced from that calculated with snow (output by default) to provide the SSRE. Depending on the spectral resolution of solar energy in each model, it would also be useful to provide the visible and near-infrared partitions of these fields:

- Net surface visible (0.2-0.7$\mu$m) irradiance calculated without snow
- Net surface near-IR (0.7-5.0$\mu$m) irradiance calculated without snow
- Mean visible surface albedo calculated without snow
- Mean near-IR surface albedo calculated without snow



Although the no-snow albedo fields are not strictly needed for the calculation of SSRE, they will complement standard albedo output from the model to facilitate convenient evaluation and the derivation of hypothetical SSRE from different (e.g., clear-sky) surface downwelling irradiance fields.

### 3.3 Timeline, data request and data availability

The first set of reference site simulations (Ref-Site) have already been carried out and are currently being analysed. The additional site simulations will be carried out in the near future. In 2018 and 2019, the global climate modelling community will be heavily involved in CMIP6. To decrease peak work load on the modelling groups while at the same time optimizing synergies with the CMIP6 activities and in particular with LS3MIP, it was decided to launch the ESM-SnowMIP simulations after the main CMIP6 activities. This has, however, the disadvantage that ESM-SnowMIP global simulation results will not

be available for analysis in time for the 6th IPCC assessment report. The tentative schedule for the realization of the various ESM-SnowMIP simulations is indicated in Table 1.

For these global land-only and coupled simulations, the request for output variables is identical to the LS3MIP data request[2] for the respective reference simulations indicated in Table 1, and the model setup will be very similar, as described in the preceding sections of this paper. This further keeps the additional workload for the ESM-SnowMIP coupled simulations to a

minimum.

The ESM-SnowMIP site simulation output is sufficiently small to be easily handled via a ftp server at one of the participating institutes (see the dedicated website at https://www.geos.ed.ac.uk/~ressery/ESM-SnowMIP.html). Gridded northern hemisphere SWE data are freely available from the National Snow and Ice Data Center (http://nsidc.org/data/NSIDC-0668). Large-scale meteorological forcing (GSWP3) for the distributed simulations, also used in LS3MIP, will be made available via the ESGF https://esgf-node.llnl.gov/search/input4mips/. To optimize synergies with

LS3MIP, ESM-SnowMIP will seek endorsement by the WCRP Working Group on Coupled Modeling, which oversees CMIP. This will allow ESM-SnowMIP output to be handled via the Earth System Grid Federation[3], i.e. the same infrastructures as the relevant CMIP6 simulations these simulations are to be compared with.

## 4 Discussion

### 4.1 Expected outcome and impact of ESM-SnowMIP

The parameterization of "cold" land-surface processes has received varying degrees of attention by climate modelling groups. In the framework of CMIP6, there is now a specific type of numeric experiments specifically designed for evaluating the land-surface components of the current-generation Earth System models. It is envisaged that these so-called LMIP

---

[2] https://www.earthsystemcog.org/projects/wip/CMIP6DataRequest
[3] https://esgf.llnl.gov/



exercises could become a central component of future CMIP editions (van den Hurk et al. 2016) as part of the so-called DECK core experiments (Eyring et al. 2016), along with separate evaluations of the atmospheric and ocean components.

The specific effort aiming at evaluating and improving the representation of snow in ESMs and in more specific dedicated snow models relates to this broader context. It is hoped that ESM-SnowMIP, in conjunction with LS3MIP, will provide a

clear determination of the current state of the art of snow modelling in the different participating communities and will spur knowledge transfer between these partially disjoint scientific communities. Snow modules in current-generation ESMs show a large range of degrees of sophistication. We expect that ESM groups who have not devoted particular effort to evaluating and improving their snow modules in the past will be able to benefit from a clear strategy for priorities of first-order snow module enhancements identified within ESM-SnowMIP. For those groups that have already put substantial effort into testing

and continuously adapting their ESM snow modules or specific snow models, ESM-SnowMIP will be an opportunity to assess past and determine future priorities for model enhancement.

The intended assessment of snow-related feedbacks on interannual and longer timescales, including a multi-model evaluation of the snow shortwave radiative effect, will hopefully help better constrain the global climate response to anthropogenic forcing and better understand regional responses, including the amplification of global warming at high

Northern latitudes.

### 4.2 Possible future extensions

Recent work on tundra snow (Domine et al. 2016) has highlighted the importance and particularity of snow metamorphic processes under very cold conditions, specifically in the presence of strong vertical temperature gradients. While wind compaction in the absence of shelter by higher vegetation can increase snow density (Sturm et al. 2001) and hence snow

conductivity, depth hoar formation induced by strong vertical temperature gradients within the snow pack (Derksen et al. 2014, 2009) can substantially reduce the conductivity (Domine et al. 2016). In ESM-SnowMIP, an effort will be made to include snow observation sites from tundra environments in the near future (e.g., Boike et al. 2017). However, it is clear that in future extensions of ESM-SnowMIP, snow on sea ice and on the polar ice sheets should also move into the focus of attention. The physical properties of snow on sea ice are linked to low accumulation rates and strong vertical temperature

gradients, its spatial heterogeneity, and its peculiar evolution in summer leading to melt ponds on sea ice due to inhibited drainage of meltwater. These are specificities that are, to our knowledge, often not taken into account in Earth System Models; however, snow on sea ice obviously concerns the sea-ice module of the coupled models and thus a slightly different community than that mobilized in this first version of ESM-SnowMIP. Assessments of snow on sea ice usually focus on, and are usually limited to, snow mass and height (e.g., Blanchard-Wrigglesworth et al. 2015; Hezel et al. 2012). However, an

extension of the ESM-SnowMIP approach based on combined small- and large-scale evaluation of snow models will





necessarily be conditioned by the availability of yet scarce observations. In this context, the MOSAIC international Arctic drift expedition[4] is projected to provide valuable new observations.

A further obvious future extension of ESM-SnowMIP should tackle snow in extreme polar environments such as Dome C on the East Antarctic Plateau or the Greenland Summit; long-term meteorological observations have already been used at these

locations for testing stand-alone snow models (Libois et al. 2015; Carmagnola et al. 2013). Perennial snow cover evolving under such extreme conditions would substantially broaden the range of snow types evaluated within our framework. This would provide stringent tests for snow models designed to simulate snow types that are extreme but far from rare, as the interior regions of the continental polar ice sheets make up an essential part of the perennial cryosphere.

ESM-SnowMIP combines model evaluation at local and global scales. While snow, and particularly the timing and intensity

of its melt season, do have important effects on basin-scale hydrology (e.g., Berghuijs et al. 2014; Barnhart et al. 2016; Fyfe et al. 2017), basin-scale processes exert a less dominant control on snow as such in terms of physical properties. Therefore intermediate scales, such as addressed in, e.g., the Rhône-Aggregation Land Surface Scheme Intercomparison Project (Boone et al. 2004), are bridged in the current phase of ESM-SnowMIP. However, terrain configuration and vegetation distribution, that is, geographical characteristics at intermediate (basin) scales, have an obvious and important effect on the

snow cover, in particular on snow cover fraction. In the current phase of ESM-SnowMIP, such links are implicitly addressed through the assessment of snow cover fraction in the prescribed SWE experiment SWE-LSM. In future phases, basin-scale properties, processes and characteristics might be addressed more explicitly. A further aspect involving unresolved scales that could be tackled in future extensions of ESM-SnowMIP is the high spatial variability of impurities deposed on snow surfaces, given the known strong impact of this deposition (Flanner et al. 2007) and model errors that can be induced by not

taking this effect into account (e.g., Clark et al. 2015).

### Acknowledgments

We acknowledge the World Climate Research Programme's Working Group on Coupled Modelling, which is responsible for CMIP, and we thank the climate modelling groups participating in CMIP5 for producing and making available their model output. For CMIP the U.S. Department of Energy's Program for Climate Model Diagnosis and Intercomparison provides

coordinating support and led development of software infrastructure in partnership with the Global Organization for Earth System Science Portals. Analysis of ESM-SnowMIP reference site simulations is supported by NERC grant NE/P011926/1 IGE and CNRM/CEN are part of LabEX OSUG@2020 (ANR10 LABX56). Simulations by the SWAP model were supported by the Russian Science Foundation (Grant 16-17-10039, where Ye.M. Gusev is the recipient).

---

[4] http://www.mosaicobservatory.org/experiments.html



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

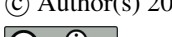



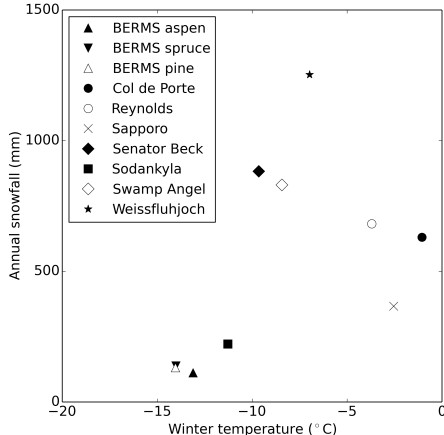

**Figure 1: Average winter (DJF) temperature and annual snowfall for the ESM-SnowMIP reference sites.**

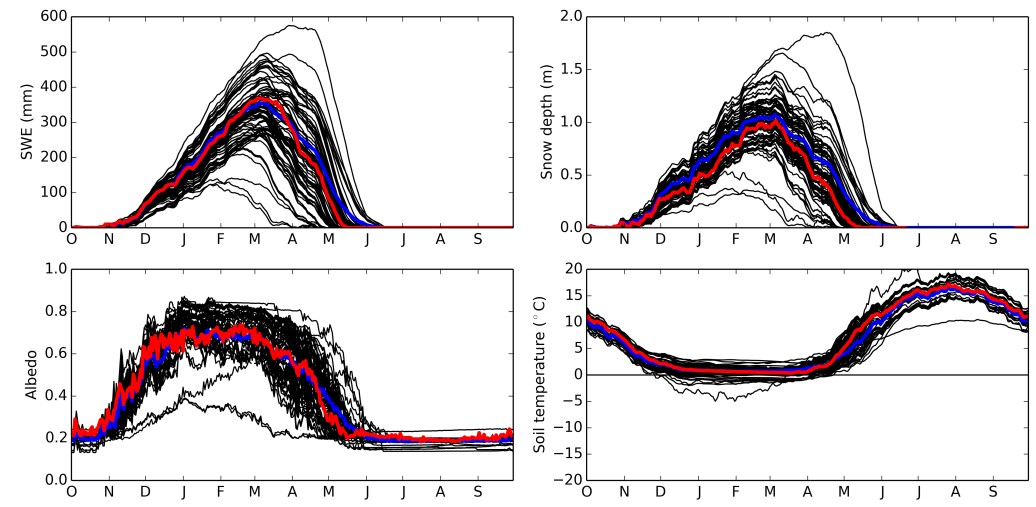

**Figure 2: Measurements (red lines), simulations (black lines) and averages of simulations (blue lines) of SWE, snow depth, albedo and soil temperature at 20 cm depth for Col de Porte, averaged over October 1994 to September 2014.**



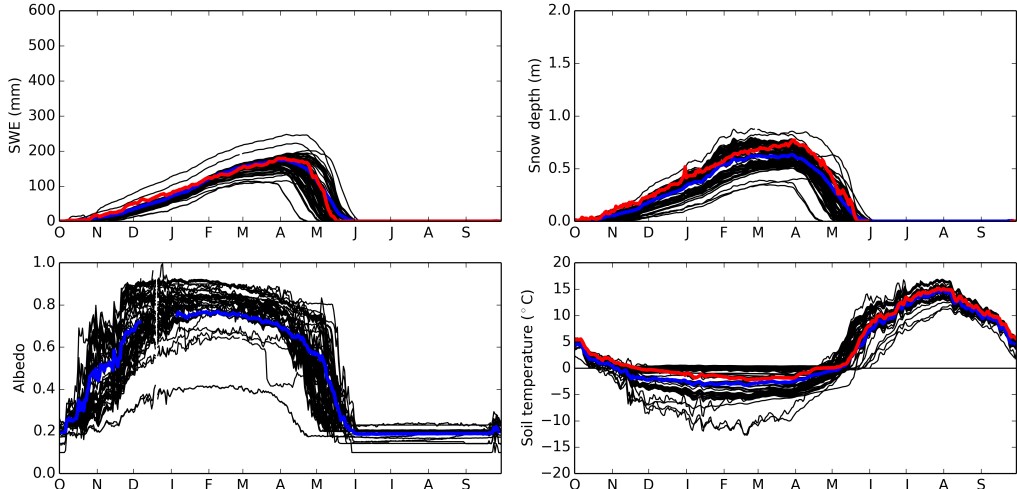

**Figure 3: As Figure 2, but for October 2007 to September 2014 at Sodankylä (albedo measurements are not available for the snow surface).**

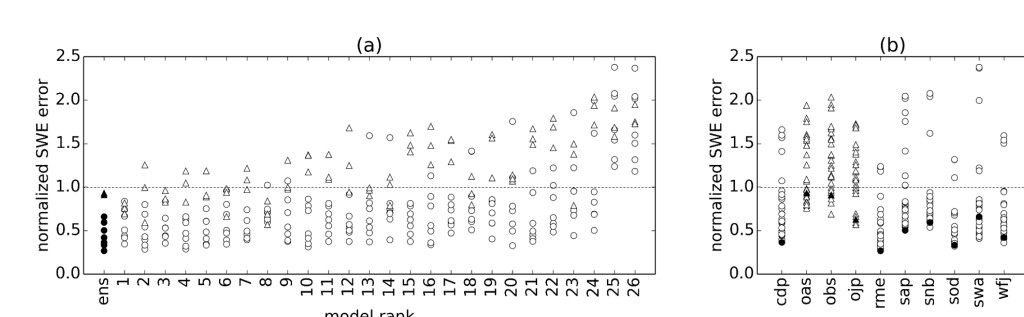

**Figure 4: Normalized root mean square SWE errors for the 26 non-ensemble models in Table 3 returning a single simulation for each site (white symbols) and the 26 model ensemble mean (black symbols), with simulations identified by circles at open sites and triangles at forested sites. (a) Models ranked according to their average error for all sites. (b) Errors for all models at each site (for**
10 **abbreviations see Figure 4).**



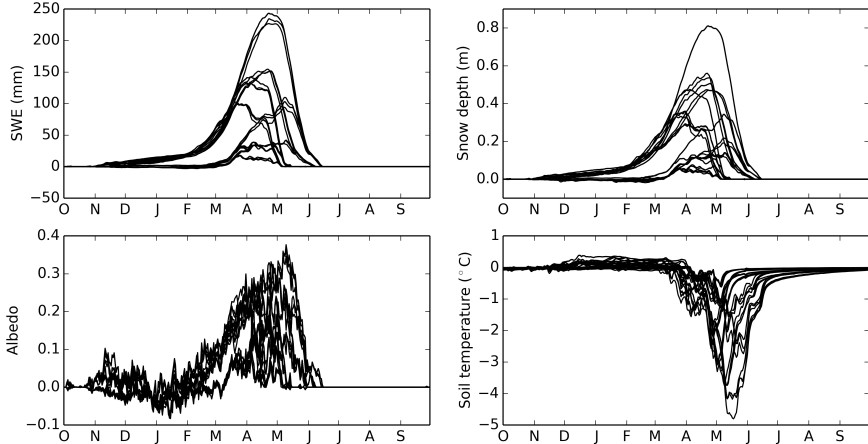

**Figure 5: Differences between FSM ensembles of fixed albedo and Correct prescription reference simulations for Col de Porte, averaged over October 1994 to September 2014.**

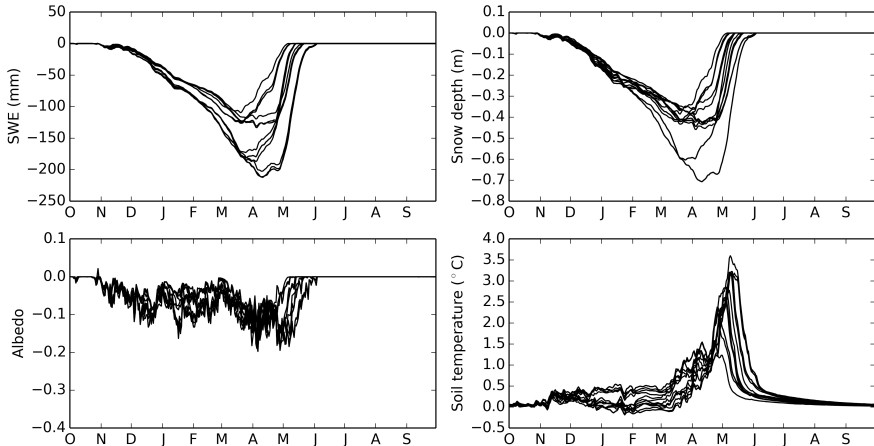

5 **Figure 6: As Figure 5, but for differences between FSM ensembles of simulations with a fixed surface exchange coefficient over snow and reference simulations.**



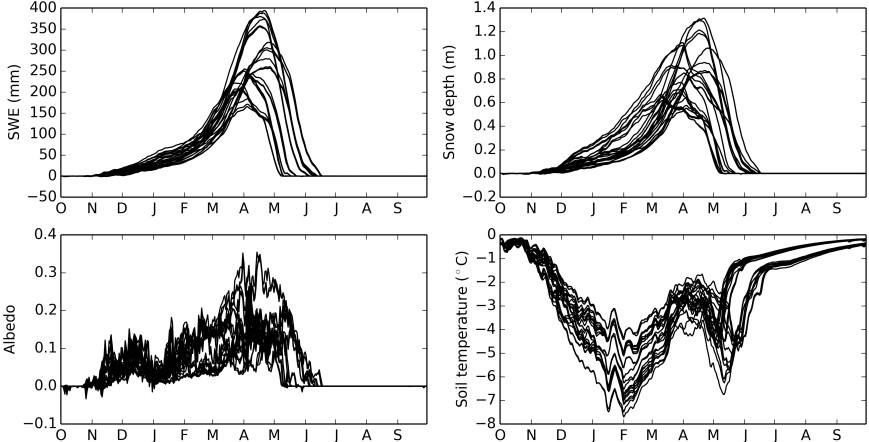

**Figure 7: As Figure 5, but for differences between FSM ensembles of simulations with very high thermal conductivity for snow and reference simulations.**

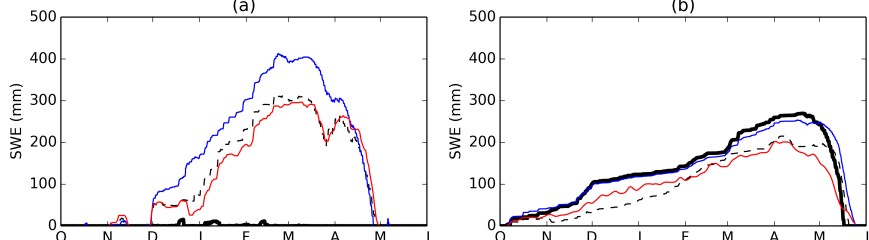

**Figure 8: Simulations with a single FSM ensemble member and in situ driving data (dashed black lines), large-scale GSWP3 driving data (solid black lines) or bias-corrected GSWP3 driving data (blue lines) compared with in situ SWE measurements (red lines) for 2009-2010 at (a) Col de Porte and (b) Sodankylä.**



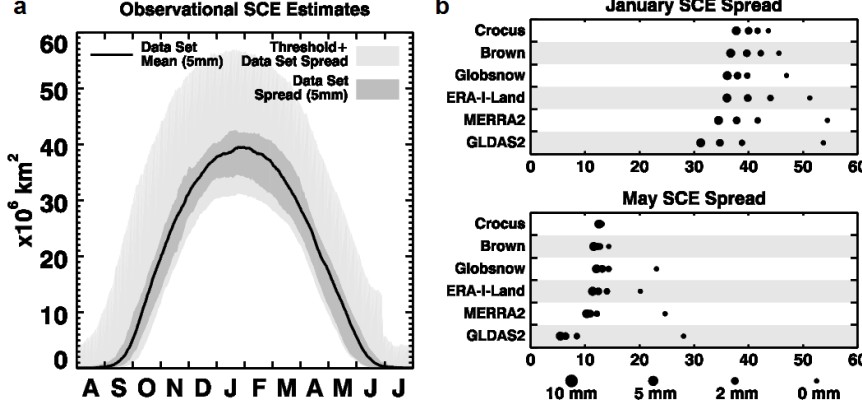

**Figure 9: Daily median and spread (5th—95th percentile) among all seven snow analyses listed above for the 1981—2010 period.**

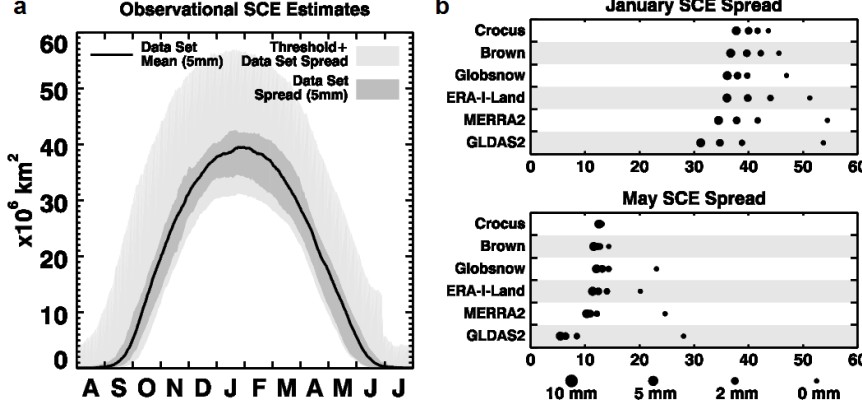

**Figure 10: (a) Daily median and spread (5th-95th percentile) among all seven snow analyses for SCE calculated using a 5mm SWE**
5 **threshold (solid curve and dark shading) and spread calculated using a range of thresholds between 0-10mm (light shading). (b) January and May SCE for four choices of thresholds.**





| Experiment name | Tier | Experiment description / design | Con-figu-ration | Start End | #Yrs per run | Ens. size | #Yrs total | Science question and/or gap addressed with this experiment | Possible synergies with other runs | Run schedule |
|---|---|---|---|---|---|---|---|---|---|---|
| **Ref-Site** | 1 | Site reference simulations | LND 1D | Variable | | | | Evaluate snow model on site scale | LS3MIP LMIP-H | 2017- |
| **FA-Site** | 2 | Site simulations, prescribed constant snow albedo | LND 1D | Variable | | | | Evaluate effect of snow albedo variations | Ref-Site | 2018- |
| **NS-Site** | 2 | Site simulations, prescribed neutral exchange coefficient | LND 1D | Variable | | | | Quantify effect of melt-induced near-surface temperature inversions | Ref-Site | 2018- |
| **NI-Site** | 2 | Site simulations, no soil insulation | LND 1D | Variable | | | | Diagnose snow soil insulation effect | Ref-Site | 2018- |
| **LSF-down-scaled-Site** | 2 | Site simulations, downscaled forcing | LND 1D | Variable | | | | Evaluate impact of downscaled gridded forcing in complex topography | Ref-Site | 2018- |
| **SWE-LSM** | 1 | Prescribed observed snow water equivalent | LND | 1980-2014 | 35 | 1 | 35 | Evaluate link between snow mass and snow fraction | Land-Hist (LS3MIP) | 2019- |
| **FA-LSM** | 2 | Land only simulation, prescribed constant snow albedo | LND | 1980-2014 | 35 | 1 | 35 | Evaluate effect of snow albedo variations | Land-Hist (LS3MIP) | 2019- |
| **NI-LSM** | 2 | Land only simulation, no soil insulation | LND | 1850-2014 | 165 | 1 | 165 | Diagnose snow soil insulation effect | Land-Hist (LS3MIP) | 2019- |
| **FLC-LSM** | 2 | Land only | LND | 1980- | 35 | 1 | 35 | Diagnose effect | Land-Hist | 2019- |





| | | simulation, prescribed common land cover | | 2014 | | | | of varying prescribed land covers | (LS3MIP) | |
|---|---|---|---|---|---|---|---|---|---|---|
| **SnowMIP-rmLC** | 1 (2) | Prescribed snow conditions 30-year running mean | LND-ATM-OC | 1980-2100 | 121 | 1 (+4) | 121 (+484) | Diagnose snow-climate feedback including ocean response | CMIP6 historical, Scenario-MIP, LFMIP-rmLC | 2019- |

**Table 1: Proposed ESM-SnowMIP simulations. Configurations are: LND 1D: Site-scale 1d (column); LND: Global land simulations with LSMs; LND-ATM-OC: Coupled land-atmosphere-ocean simulations.**

| Site | Latitude | Longitude | Elevation | Period | Type | Reference |
|---|---|---|---|---|---|---|
| BERMS[5] Old Aspen, Canada (oas) | 53.63°N | 106.20°W | 600 m | 1997-2010 | Boreal | Bartlett et al. (2006) |
| BERMS Old Black Spruce, Canada (obs) | 53.99°N | 105.12°W | 629 m | 1997-2010 | Boreal | Bartlett et al. (2006) |
| BERMS Old Jack Pine, Canada (ojp) | 53.92°N | 104.69°W | 579 m | 1997-2010 | Boreal | Bartlett et al. (2006) |
| Col de Porte, France (cdp) | 45.30°N | 5.77°E | 1325 m | 1994-2014 | Alpine | Morin et al. (2012) |
| Reynolds Mountain East, USA (rme) | 43.06°N | 116.75°W | 2060 m | 1988-2008 | Alpine | Reba et al. (2011) |
| Sapporo, Japan (sap) | 43.08°N | 141.34°E | 15 m | 2005-2015 | Maritime | Niwano et al. (2012) |
| Senator Beck, USA (snb) | 37.91°N | 107.73°W | 3714 m | 2005-2015 | Alpine | Landry et al. (2014) |
| Sodankylä, Finland (sod) | 67.37°N | 26.63°E | 179 m | 2007-2014 | Arctic | Essery et al. (2016) |
| Swamp Angel, USA (swa) | 37.91°N | 107.71°W | 3371 m | 2005-2015 | Alpine | Landry et al. (2014) |
| Weissfluhjoch, Switzerland (wfj) | 46.83°N | 9.81°E | 2540 m | 1996-2016 | Alpine | WSL (2017) |

5   **Table 2: ESM-SnowMIP reference sites with abbreviations used in Figure 4.**

---

[5] BERMS = Boreal Ecosystem Research and Monitoring Sites





| Model | Type |
|---|---|
| BCC_AVIM | LSS in BCC-ESM |
| CABLE | LSS in ACCESS |
| CRHM | hydrological model |
| CLASS | LSS in CanESM |
| CLM5 | LSS in CESM |
| CoLM | LSS in BNU-ESM and CAS-ESM |
| Crocus | snow physics model |
| ecearth | LSS in EC-EARTH |
| ESCIMO | snow surface energy balance model |
| ESCROC | multi-physics snow model (35-member ensemble) |
| FSM | multi-physics snow model (32-member ensemble) |
| htessel | LSS of ECMWF operational forecasting system |
| htesselML | LSS of ECMWF forecasting system (research) |
| ISBA-ES | LSS in CNRM-CM |
| ISBA-MEB | LSS in CNRM-CM |
| JSBACH | LSS in MPI-ESM |
| JSBACH3_PF | LSS in MPI-ESM |
| JULES | LSS in UKESM |
| MATSIRO | LSS in MIROC |
| MOSES | LSS in HadCM3 |
| ORCHIDEE-E | LSS in IPSL-CM |
| ORCHIDEE-I | LSS in IPSL-CM |
| RUC | LSS in NOAA/NCEP operational forecasting systems |
| SMAP | snow physics model |
| SNOWPACK | snow physics model |
| SPONSOR | hydrological model |
| SWAP | LSS |
| VEG3D | soil and vegetation model |

**Table 3: Models performing ESM-SnowMIP reference site simulations.**