# Peer review of "ESM-SnowMIP: Assessing models and quantifying snow-related climate feedbacks"

_Geoscientific Model Development, 2018_

## Referee Comment (RC1) · Anonymous Referee #1 · 31 Aug 2018

doi.org/10.5194/gmd-2018-153

ESM-SnowMIP: Assessing models and quantifying snow-related climate feedbacks

Gerhard Krinner et al.

This is a slightly unusual paper in that it presents a comprehensive overview of a planned model intercomparison, with a few early results, rather than a critically assessed specific model development. However, what is being proposed is a very significant international effort to understand the representation of snow, and its interactions with the atmosphere and substrate, in earth system models. Consequently, it is worthy of presentation at this stage and is aligned to a suite of other papers in GMD that describe model experiments either directly part of, or running in parallel to, CMIP6. This

paper is not just an advert for what is to come, which in itself would not be suitable for publication. In particular, there are some interesting early results from the tier 1 'Ref-Site' experiment. The paper is very well written and easy for a wider audience to understand, providing a valuable reference document for further publications as the results of each of the ten experiments materialize. Consequently, I only have a few comments that may improve description of the experimental framework, interpretation of results currently presented, as well as correcting a few minor typos in the current manuscript:

1. In the 'Objectives and Rationale' section can you specifically say what lessons have been learned from previous MIPs (especially SnowMIP 1 &2)? MIPs are often criticized (for good reason), for not providing clear and incisive direction for the way forward in model development. Rather they too often fall back on the conclusions that there are big spreads in model outcomes from which we cannot untangle the relative impact of uncertainties in input data, model parameters or model structure (to paraphrase an insightful pers com from Drew Slater). The relatively recent development of FSM and ES-CROC give the hope that this common failing will not be replicated in ESM-SnowMIP. However, in this section I strongly recommend that the authors explicitly say what has been unsatisfying in past snow related MIPs, what potential solutions are available, and specifically describe how ESM-SnowMIP will avoid these potential pitfalls.

2. A clear aim is to identify an 'optimum degree of complexity' (pg5, ln 4). A definition of model complexity and, more importantly, a useful and workable metric to quantify complexity requires clear guidance in this paper. This is a topic which is often broadly discussed, and which people can have a general feel for, but it rarely has a satisfying quantifiable definition and it varies with model purpose. As it seems like this will be mainstream to evaluations in ESM-SnowMIP experiments it is important this is clearly clarified here.

3. Figure 4 (and Pg7,ln 10) is a good analytical result. However, even in a paper

of this introductory nature, there needs to be some critical comment of these results. Saying 'A couple of models do well, and a couple do poorly' is wholly inadequate. I'm not recommending any 'name and shame' approach, but can you at least name some of the models that do consistently well (e.g. the first ranked model)? Surely they are doing well for a reason that we can learn from? At the very least, or in combination with naming high performing models, is there potential to highlight models anonymously by Type (see Table 3). This may go some way to addressing the complexity issue (see previous point).

Pg2 ln 25: Citations are neither listed chronologically or alphabetically. This needs consistency throughout the manuscript.

Pg2 ln 28: Consider removing 'Thermal' at the start of the sentence to help improve readability; 'thermal is currently repeated three times within that sentence.

Pg3 ln4: Consider citing Sturm et al. (2017) (doi:10.1002/2017WR020840) in addition to Fyfe et al. (2017).

Pg3, ln13: consider adding 'microstructure' to the list of physical properties. Potentially use Räisänen et al. (2017) (doi:10.5194/tc-11-2919-2017) as a citation to corroborate its inclusion.

Pg3,ln19: Be specific about the important processes you are referring to here. I'm presuming it's thermal conductivity / snow microstructure as you cite Domine et al. (2016). If so, please say so specifically and highlight the suite of processes you deem most important to consider here.

Pg3, ln31: replace 'of' with 'for'.

Pg4, ln25: cite where this wealth of new large-scale observational data sets are described.

Pg5, ln27: remove comma after 'and'.

[Figure]

Pg7, ln 8: I like this evaluation metric. However, there is slight ambiguity, e.g. is the standard deviation of measured SWE at all sites? Please consider adding an equation here to explicitly define how this is calculated.

Pg7, ln 21: replace 'validation' with 'validation of'.

Pg7, ln 26: add citation(s) to support the difficulty to untangle feedback effects.

Pg 8, ln 8: will this experiment take place? If so leave it in, if there is doubt, consider removing this sentence.

Pg8, ln 24: why use such a large thermal conductivity, which is two orders of magnitude than physically measured by Sturm et al. (1997)? While the exact value is unlikely to be critical, this needs a better justification.

Pg9, ln 22: Author name missing in the citation – presume it is Mudryk et al. (2015)?

Pg10, ln 13: Provide a citation(s) for the satellite based observations of climatological SWE.

Pg11, ln 10: section number missing in cross-reference, currently says 'section 0'. Could section 3.2.1. just be cross-referenced here to prevent repetition?

Pg15, ln 10 & Table 1: As the schedule is tentative, could you just say 'post CMIP6' here and remove the column 'Run-Schedule' in Table 1, which I think is superfluous.

Pg16, ln 17: remove 'and particularly'.

Pg16, ln23: replace 'move into the' with 'become a'.

Pg17, ln18: replace 'deposed' with 'deposited'.

Pg17,ln19: impact on what – I presume albedo, but please state this here.

Pg17, 28: did you mean to say 'Ye.M.Gusev' rather than 'Y.M.Gusev'?

Figure 5: Caption – remove 'correct prescription', and just keep 'reference simulation'

[Figure]

as per main text body.

Table 3: Must add in citation(s) that describe each model so readers have a point of reference.

[Figure]

---

## Referee Comment (RC2) · Anonymous Referee #2 · 20 Sep 2018

General comments

ESM-SnowMIP: Assessing models and quantifying snow-related feedbacks Krinner et al. 2018

This is the protocol description paper for ESM-SnowMIP, an international effort to systematically evaluate and compare different snow models across climate models. Next to listing and describing the experiments a lot of background information is provided as well as initial results from some first stand-alone point simulations, which aids in understanding the purpose of the entire exercise and makes it more enjoyable to read.

It is explained multiple times in the paper that ESM-SnowMIP builts upon the framework of one of the official CMIP6 efforts, namely LS3MIP. Indeed, it can be viewed as an

unofficial extension of that with more gridded sensitivity and single-point experiments geared specifically towards terrestrial snow, although in the final section of the paper the authors hint at including snow on sea ice in the future as well.

The manuscript is well structured and the level of detail is fine. However, the phrasing could be more accurate, given the fact this is a protocol paper. E.g. on P11 L9 it reads: "While a number of snow analyses are available to serve as prescribed SWE, we recommend the Mudryk et al. (2015) combined climatology." This can be interpreted as if it were that the decision on the dataset had still not been made. I suggest that the authors use more precise words like "propose" or "selected" in these circumstances.

Another problem is that sentences are sometimes quite long and therefore incomprehensible, e.g. P5 L4: "Global snow simulations are subject to uncertainty in the meteorological data used to drive models (whether provided by bias corrected reanalyses as in LS3MIP offline land model experiments (van den Hurk et al. 2016) or by coupling with atmospheric models as in CMIP6), global products providing vegetation and soil characteristics for model parameters are often contradictory, and global observations of snow properties for evaluation of models (e.g., for snow density and thermal conductivity) are limited." This is an extreme example but let's say that generally the readability could be improved by shortening and structuring sentences better, and removing parentheses where possible. This is a general comment that applies to the entire manuscript, and in the Specific Comments below I highlighted some more examples.

To conclude, this paper is definitely worthy of publication after the readability is improved and outstanding questions are resolved (see below).

Specific Comments

P1 L1 (title): suggest to change "assessing models" into "assessing global snow models"

P2 L15-18: it should be made more clear that ESM-SnowMIP is no official part of CMIP6, but will run parallel to it

P2 L21: suggest to replace "Northern" by "northern hemisphere" and "continental" by "terrestrial"

P2 L22: another important role of snow in the climate system is its ability to store / buffer large amounts of freshwater.

P2 L22 and beyond: "The former" indicates that there will be "a latter", but that never comes. Instead, an enumeration follows of all interactions that snow has on the climate system, with a much wider spread than signaled to us in L21-L22. Suggest to rewrite this part of the introduction to make clear we are going to read a long enumeration.

P2 L24: suggest "main driver" rather than "thought to contribute"

P2 L29: order of citations seems random. Suggest chronological order. Applies to whole document.

P3 L2: water and energy availability, suggest to add citation Rhoades et al., 2015, J. App. Met. Clim.

P3 L13: suggest to add "prognostic albedo"

P3 L28: add reference Slater 2017 to list of thermal properties of snowpack

P3 L33: on top of that, imperfect meteorological boundary conditions ?

P4 L12: suggest to remove "and specifically to (..) which is part of CMIP6", because it is basically said again in the next sentence and the sentence is already very long.

P4 L24: "see the section on reference site simulations" be more specific, which number?

P4 L25: "wealth of new large-scale observational data" -> please give an example or reference

P4 L30: what is meant with the term "mutually consistent"?

P4 L31: term AMIP is used but not explained – suggest to replace with "atmosphere only" Also, it is not clear why AMIP is mentioned at all. This paper does not use it?

P5 L4: suggest to rewrite into enumeration to improve readability. "We aim to (1) identify the optimum (. . .); (2) identify previously (. . .); and (3) identify feasible (..)"

P5 L9: second -> fourth? The authors need to work this part because the enumerations are not clear.

P5 L10: remove reference "van den Hurk 2016" to improve readability. Applies to all later occurrences in same context.

P5 L13: suggest to rewrite "To this end, diagnoses of (. . .)" and delete "complete this aspect of ESM-SnowMIP"

P5 L21: suggest to replace "supposed" by "mandatory" and rewrite second part of sentence using the word "exception".

P5 L28: suggest to rewrite this sentence into two sentences. "First, we describe single-point experiments, some of which have already been carried out and analyzed. Then, the spatially distributed experiments are described, that naturally only will be carried out by a subset (. . .)"

P5 L21: suggest to rewrite to "the planned fully coupled ESM experiments".

P6 L4 – L10 & L20-L22 : these sentences fit better in Section 3 – Experimental Design, explaining why the gridded simulations have been augmented by single-point simulations. Also, suggest to add a sentence saying that there are basically two groups of snow models participating: those doing the gridded AND point experiments, and those only doing the latter. It is implicitly clear from Section 3 but would be good to make explicit.

P6 L18: please quantify number of sites and years

P6 L20: Distinction between climate models and ESMs is superfluous? Nowadays, they basically mean the same thing, since the usual definition of ESM is a climate model with an active carbon cycle.

P6 L23: suggest to replace "sophisticated snow physics models" by "standalone snow models" since the word sophisticated is subjective.

P6 L26: at what frequency are these measurements typically available?

P7 L1: This phrasing is not very formal. Do you mean to say that the onset of melt is both under- and overestimated by models?

P7 L4: is this because there is no snow, thus no insulating effect?

P7 L15-16: suggest to split into two sentences. Further, I would say that soil temperature is also dependent on the amount of meltwater refreezing and the refreezing depth (see e.g. Van Kampenhout et al., 2017)

P7 L26: Suggest "metric" rather than "variable"

P7 L28: Please add reference for the statement "which approximates the CMIP5 multi-model mean peak snow albedo" if you have one.

P8 L3-L4: suggest to change "surface layer" to "atmospheric boundary layer" to avoid ambiguity with the top snow layer.

P8, L11: Please explain briefly why the ensemble members react differently, for people that don't know FSM.

P8 L27-29: Unclear, please rewrite.

P8 L32: acronym GSWP3 is first used but not defined nor referenced

P9 L2: suggest to avoid the words "overestimate" and "underestimated" because these have negative connotations. What you mean to say is that there is a differences between the grid cell means and the measurement site because they differ in elevation.

P9 L3: is this coincidence, or is Sodankylä located in flat terrain? Probably good to mention.

P9 L5 – L10: I would not call bias-correction a form of downscaling. A (statistical) downscaling procedure would use the climate data as is, then projecting that onto a high resolution topography using lapse rates and possibly repartitioning of precip.

P9 L17: SCF first used but not defined

P10 L3: unclear what is meant with "fully characterized bias and error"

P10 L9: SCE first used but not defined

P10 L9 – L14: can we do even better by defining a unique threshold for each of the different snow products? Was this tested?

P10 L16 – L22: Unclear; make more clear that LS3MIP is used as a baseline or control experiment and that ESM-SnowMIP adds sensitivity experiments on top of that. Reference Kim 2018 should be given earlier, when GSWP3 is introduced.

P10 L25: and prescribed SCF? If not, mention that this is left up to the model (e.g. on P11 L4)

L11 L2: "by less than 10% or so" : too vague

P11, L9: on P9 L19 you wrote that "we have developed a blended dataset" which suggested that it was specifically developed for this MIP. This contrasts with the fact you now use the reference Mudryk 2015. Indicate if and how the blended dataset used is different from Mudryk 2015 and avoid inconsistencies.

P11 L10: section number missing

P11 L28: Aren't all ESM-SnowMIP gridded experiments?

P12 L1 - L2: are these observational data for evaluation already known? If yes, mention them.

P12 L10: I don't see how the active layer depth can be diagnosed from the lowermost soil layer depth, which may vary across models.

P12 L27: "ESA 2017" citation is strange and refers to the user guide. Suggest to include these references instead: Poulter 2015 doi:10.5194/gmd-8-2315-2015

P12 L34: clarify what CCI 200 means

P13, L9 – L18: Please clarify this part. In particular unclear is this sentence "This LS3MIP experiment uses (...) and prescribes these in the LFMIP-rmLC experiment." What is meant with "scenario simulation" (mind that scenario = future in CMIP terminology) and "context of a transient run".

P13 L18: replace "we suggest carrying out" with "we propose". See earlier comment regarding protocol document.

P13 L19: Unclear whether the run being proposed here is SnowMIP-rmLC? Make explicit.

P13 L26: see earlier comment about the word "scenario"

P13 L27: remove sentence; has just been explained.

P14: this page breaks the general style of the manuscript by using bullet points extensively.

P14 L27 – L32: suggest to move this to the end of the section, P15 L3.

P16, L16: it should be made more clear whether the possible future extensions are intended for a follow-up project (ESM-SnowMIP Phase 2 or whatever) or as an integral part of the current effort.

P16, L24: and bottom heat flux, presence of salt

P18 Many citations are not up to date and contain either the word "Received" (e.g. L7) or they point to a discussions paper (e.g. L22). Next to that the use of URLs is

not consistent, e.g. L25 contains an URL that basically repeats the DOI. Suggest to remove all URLs and stick to DOI. Whole reference list needs a cleanup like this.

P21, L20: this work is not available under any DOI so reference should be removed.

P25, Figure 1: over what period was the average computed?

P27 Figure 5 – 7: suggest to replace "ensembles" with "ensemble members" and mention in each case how many ensemble members are present in the graph.

P27, Figure 5: What is meant with "Correct prescription"? Further, the word "period" is missing.

P29, Figure 10: The 7th dataset, MERRA, appears to be missing from the graph

P31, Table 2: suggest to put site abbreviations / tags in separate column for quick reference

Technical corrections

P2 L12 and L15: choose either "modelling" or "modeling" and be consistent throughout manuscript.

P2 L13: suggest to move "against local and global observations" to the end of the sentence

P2 L14: suggest to replace "at identifying" by "to identify" and remove "snow"

P3 L19: suggest to move "particularly in cold conditions" to end of the sentence to improve readability

P3 L31: suggest to replace "a large room of improvement" with "room for improvement"

P4 L3: PILPS is an acronym and should be capitalized.

P4 L3: "(...) Phase 2d (Slater 2001), Phase 2e (Bowling et al 2003), as well as SnowMIP Phase 1 (..) "

[Figure]

P4 L12: CMIP6 acronym has already been defined in abstract; no need to repeat.

P4 L27: suggest to replace "to profit" by "take advantage"

P5 L17 and L19: suggest to remove superfluous "rapid" in both places.

P5 L24: the word "Tier" should be capitalized, as is done later in the paper

P6 L10: suggest to replace "These process-based studies have been enabled" by "The single-point experiments are enabled"

P6 L13: suggest to replace "and aerosol" with "or aerosol"

P6 L15: Arctic should be capitalized

P6 L26: suggest to rewrite to "Snow water equivalent (SWE), depth measurements, and hence bulk snow density are available for all of the reference sites. Several sites also (. . .)" to improve readability

P6 L30 : suggest to rewrite "close to the model ensemble means"

P8 L17: rewrite to "(. . .) has a major climatic control on (. . .)"

P8 L18: rewrite to "(..) that is spatially and temporally variable and often (. . .)"

P8 L19: replace "attributed" with "assumed"

P9 L1: rewrite to "with an average elevation of 870 m".

P9 L13: suggest to rewrite to "identifying unique priorities for development of each of the participating models." thus removing repetition.

P9 L19: "dataset" is one word

P9 L19 & L24 & beyond: suggest to rephrase "snow analyses" to "snow products" which for me better covers the meaning. Also see caption of Figure 9.

P9 L22: reference is missing

P10 L14: Figure 10a

P11, L17: suggest to replace "erroneous" with "biased"

P11, L24: rewrite to "over ice sheets"

P12, L5: Suggest to rewrite to "this simulation will otherwise have" & then delete the part that reads "except (. . .) conductivity"

P13, L10: remove superfluous "combined"

P16 L7 : suggest to rephrase "large range of degrees of sophistication" to "a wide spread in their degree of complexity".

P16, L14: suggest to replace "including" by "such as"

P17, L1: suggest to rewrite "yet scarce observations" to "high-quality observations"

P17, L10: replace 'do' by 'does'

---

## Author Comment (AC1) · 14 Nov 2018

**1  Reviewer 1**

We thank the reviewer for his thoughtful and detailed comments. In the following, we only reply explicitly to comments that require a detailed response. Therefore, the comments that were really trivial to address (such as: *Pg17, ln18: replace 'deposed' with 'deposited'*) were taken into account and the paper was revised accordingly, but they are not explicitly mentioned in the following.

**Reviewer comment**:

1. In the 'Objectives and Rationale' section can you specifically say what lessons

have been learned from previous MIPs (especially SnowMIP 1 &2)? MIPs are often criticized (for good reason), for not providing clear and incisive direction for the way forward in model development. Rather they too often fall back on the conclusions that there are big spreads in model outcomes from which we cannot untangle the relative impact of uncertainties in input data, model parameters or model structure (to paraphrase an insightful pers com from Drew Slater). The relatively recent development of FSM and ES-CROC give the hope that this common failing will not be replicated in ESMSnowMIP. However, in this section I strongly recommend that the authors explicitly say what has been unsatisfying in past snow related MIPs, what potential solutions are available, and specifically describe how ESM-SnowMIP will avoid these potential pitfalls.

Reply:

We added the following paragraph at the beginning of the section: "Common conclusions emerging from previous snow model intercomparisons in PILPS and SnowMIP (Slater et al. 2001; Nijssen et al. 2003; Essery et al. 2009) are that there are large differences between models and that these differences are largest at warmer sites, in warmer winters, and during spring snowmelt. Little insight has been gained into how to reduce this model uncertainty, but it is precisely in the warmer regions that current snow cover is most "at risk" from climate warming (Nolin and Daly 2006) and where most confidence in projections is required. There has also been a lack of connection between intercomparisons at site scales, where detailed analyses of snow processes are possible, and intercomparisons at global scales, where projections of changes in snow cover are required."

**Reviewer comment**:

A clear aim is to identify an 'optimum degree of complexity' (pg5, ln 4). A definition of model complexity and, more importantly, a useful and workable metric to quantify complexity requires clear guidance in this paper. This is a topic which is often broadly

discussed, and which people can have a general feel for, but it rarely has a satisfying quantifiable definition and it varies with model purpose. As it seems like this will be mainstream to evaluations in ESM-SnowMIP experiments it is important this is clearly clarified here.

Reply:

We doubt that there can be a unique metric of complexity because a metric would quantify, for example, the number of parameters used in the model or the number of processes represented, without telling anything about the relevance of those processes for the purpose. However, the reviewer is perfectly right in asking for a more specific roadmap towards our aim of identifying the "optimum degree of complexity". The novelty of our approach is the simultaneous use of site-scale and global simulations and a large diversity of models, including multi-physics models. We therefore add the following sentences in order to try to make this point clearer: "It is hoped that the conjunction of global simulations with long site simulations, including sensitivity tests with simplified parameterizations (e.g., fixed albedo), and the systematic comparison with multi-physics models will provide insights into what the optimum degree of complexity for the intended applications of the various model types is. Specifically, beyond the minimum number of vertical levels required, we aim at getting a better idea about how finely the time evolution of fundamental physical properties of the snowpack (albedo, density, conductivity, liquid water content) and interactions with vegetation need to be represented particularly in climate models in order to be able to correctly simulate the most climate-relevant snow-related variables (snow fraction, thermal insulation of the underlying soil)."

**Reviewer comment**:

3. Figure 4 (and Pg7,ln 10) is a good analytical result. However, even in a paper of this introductory nature, there needs to be some critical comment of these results. Saying 'A couple of models do well, and a couple do poorly' is wholly inadequate. I'm not

recommending any 'name and shame' approach, but can you at least name some

of the models that do consistently well (e.g. the first ranked model)? Surely they are doing well for a reason that we can learn from? At the very least, or in combination with naming high performing models, is there potential to highlight models anonymously by Type (see Table 3). This may go some way to addressing the complexity issue (see previous point).

Reply:

Figure 4 is provided as an illustration of range in model performance, but it would be premature to discuss why some models perform more consistently than others or to identify the highest ranked models, and there is no clear pattern related to model type. There are many metrics that can be used for evaluating snow model performance and, with many sites and many years in the simulations, there are many ways in which models could be ranked according to each metric. The ranking of the models depends on these choices. Model evaluation is not the aim of this paper, but it will be thoroughly addressed in a forthcoming paper. However, we added the following sentence at the end of secction 3.1.2.: "The only individual models to have normalized errors less than 1 for all sites are the Crocus snow physics model and the htessel and SWAP land surface schemes, which have very different complexity. There will be a thorough evaluation of model performance in a forthcoming paper."

**Reviewer comment**:

Pg2 ln 25: Citations are neither listed chronologically or alphabetically. This needs consistency throughout the manuscript.

Reply:

In all instances of multiple citations at the same place, these are now ordered chronologically.

**Reviewer comment**:

Pg3,ln19: Be specific about the important processes you are referring to here. I'm presuming it's thermal conductivity / snow microstructure as you cite Domine et al. (2016). If so, please say so specifically and highlight the suite of processes you deem most important to consider here.

Reply:

We reformulated the statement to be more specific about relevant processes. The start of this paragraph now reads: "Particularly in very cold conditions, it is clear that some important physical processes affecting snow are not captured even by the most detailed physically-based snow models, for example the appearance of inverted density gradients due to water vapour fluxes (Domine et al. 2016; Gouttevin et al. 2018). A further important and rarely represented process is wind-blown snow and its sublimation (Pomeroy and Jones 1996)."

**Reviewer comment**:

Pg4, ln25: cite where this wealth of new large-scale observational data sets are described.

Reply:

The datasets are described and cited in section 3.2.1. We therefore reformulate the sentence and refer to that section: "Similarly, on the global scale, a wealth of new northern hemisphere datasets based on advanced remote-sensing techniques and land surface models driven by reanalysis allows for more meaningful evaluations than has been possible in the past (see section 3.2.1)."

**Reviewer comment**:

Pg7, ln 8: I like this evaluation metric. However, there is slight ambiguity, e.g. is the standard deviation of measured SWE at all sites? Please consider adding an equation here to explicitly define how this is calculated.

Reply:

We clarified this sentence by adding the relevant information (here in italics): "Figure 4 shows one example, in which root mean squared errors in simulated SWE have been calculated for each model at each site and normalized by the standard deviation of measured SWE *at the given site* for comparison between sites."

**Reviewer comment**:

Pg7, ln 26: add citation(s) to support the difficulty to untangle feedback effects.

Reply:

Here we now cite Qu and Hall (2014) and Mudryk et al. (2017), which show links between albedo and warming: "Snowmelt timing is a critical climatic metric that is often incorrectly simulated by climate and dedicated snow models, but it is difficult to untangle the effects of the simulation of snow albedo from other processes because of the strong feedbacks involved (Qu and Hall 2014; Mudryk et al. 2017)."

**Reviewer comment**:

Pg8, ln 24: why use such a large thermal conductivity, which is two orders of magnitude than physically measured by Sturm et al. (1997)? While the exact value is unlikely to be critical, this needs a better justification.

Reply:

As we had written in this paragraph, the aim was to identify a value of thermal conductivity that did not compromise numerical stability and, at the same time, led to vanishing thermal insulation by the snowpack. We therefore add the following sentence: "This value of thermal conductivity led to vanishing temperature gradients across the snowpack and was therefore deemed high enough without compromising numerical stability."

**Reviewer comment**:

Pg10, ln 13: Provide a citation(s) for the satellite based observations of climatological SWE.

Reply:

We now refer to Robinson et al. (2014): Robinson, D. A., D. K. Hall, and T. L. Mote, 2014: MEaSUREs Northern Hemisphere Terrestrial Snow Cover Extent Daily 25km EASE-Grid 2.0, Version 1.

**Reviewer comment**:

Table 3: Must add in citation(s) that describe each model so readers have a point of reference.

Reply:

Done, although we had hoped that nobody would ask us to do that.

**2 Reviewer 2**

We also thank the second reviewer for his thoughtful and detailed comments. Again, we only reply explicitly to comments that require a detailed response. Therefore, comments that were trivial to address (such as: *P2 L21: suggest to replace "Northern" by "northern hemisphere" and "continental" by "terrestrial"*) were taken into account and the paper was revised accordingly, but such comments are not explicitly mentioned in the following.

**Reviewer comment**:

P1 L1 (title): suggest to change "assessing models" into "assessing global snow models"

Reply:

Done. We changed the title to "ESM-SnowMIP: Assessing snow models and quantifying snow-related climate feedbacks".

**Reviewer comment**:

P2 L15-18: it should be made more clear that ESM-SnowMIP is no official part of CMIP6, but will run parallel to it.

Reply:

We now write: "Although it is not part of the 6th phase of the Coupled Model Intercomparison Project (CMIP6), ESM-SnowMIP is tightly linked to the CMIP6 endorsed Land Surface, Snow and Soil Moisture Model Intercomparison (LS3MIP)."

**Reviewer comment**:

P2 L22: another important role of snow in the climate system is its ability to store /buffer large amounts of freshwater.

Reply:

Yes, that's why we had written: "Linked to its effect on soil humidity, snow has an obvious and profound impact on water availability in snow-dominated regions (Barnett et al. 2005), and large potential economic impacts of snowpack decrease in a warming climate can be expected regionally (e.g., Fyfe et al. 2017; Sturm et al. 2017)."

**Reviewer comment**:

P2 L22 and beyond: "The former" indicates that there will be "a latter", but that never comes. Instead, an enumeration follows of all interactions that snow has on the climate system, with a much wider spread than signaled to us in L21-L22. Suggest to rewrite this part of the introduction to make clear we are going to read a long enumeration.

Reply:

OK, we rewrote this part of the introduction to make it easier to read.

**Reviewer comment**:

P2 L29: order of citations seems random. Suggest chronological order. Applies to whole document.

Reply:

Reviewer 1 had the same remark. We now use chronological order throughout.

**Reviewer comment**:

P3 L33: on top of that, imperfect meteorological boundary conditions ?

Reply:

Yes, we now write: "Additional uncertainty in snow modelling often comes from imperfect meteorological driving data (e.g., Raleigh et al. 2015; Schlögl et al. 2016)."

**Reviewer comment**:

P4 L12: suggest to remove "and specifically to (..) which is part of CMIP6", because it is basically said again in the next sentence and the sentence is already very long.

Reply:

Done, thank you.

**Reviewer comment**:

P4 L24: "see the section on reference site simulations" be more specific, which number?

Reply:

We added the relevant section numbers: "The availability of longer-term high-quality observations at a larger range of sites than in previous intercomparison exercises provides the opportunity for a more comprehensive assessment of the current modelling capacity in different climate settings (see section 3.1). Similarly, on the global scale, a wealth of new large-scale observational datasets based on advanced remote-sensing techniques allows for more meaningful evaluations than has been possible in the past (see section 3.2.1)."

**Reviewer comment**:

P4 L25: "wealth of new large-scale observational data" -> please give an example or reference

Reply:

This point was also made by reviewer 1. We now refer to the relevant section where these papers are cited: "Similarly, on the global scale, a wealth of new large-scale observational datasets based on advanced remote-sensing techniques allows for more meaningful evaluations than has been possible in the past (see section 3.2.1)."

**Reviewer comment**:

P4 L30: what is meant with the term "mutually consistent"?

Reply:

We reformulated the sentence to make this clear: "CMIP6 provides the opportunity to evaluate the representation of the historical evolution of seasonal snow in global simulations with varying degrees of freedom, ranging, for a given model, from global coupled ocean-atmosphere simulations to atmosphere-only (AMIP) climate simulations with prescribed oceanic boundary conditions (Gates 1992) to land-surface only simulations (LMIP) forced by observationally-based meteorological data (van den Hurk et al. 2016)."

**Reviewer comment**:

P4 L31: term AMIP is used but not explained - suggest to replace with "atmosphere only" Also, it is not clear why AMIP is mentioned at all. This paper does not use it?

Reply:

We now define this term.

**Reviewer comment**:

P5 L4: suggest to rewrite into enumeration to improve readability. "We aim to (1) identify the optimum (: : :); (2) identify previously (: : :); and (3) identify feasible (..)"

Reply:

We followed this suggestion. The sentence now reads: "Concerning this first major objective of model evaluation and improvement, we aim (1) at identifying the optimum degree of complexity required and sufficient in global models to simulate snow-related processes satisfyingly on large scales, (2) at identifying previously unrecognized weaknesses in these models and (3) at identifying feasible ways to correct these by including relevant processes and setting model parameters judiciously."

[Figure]

**Reviewer comment**:

P5 L9: second -> fourth? The authors need to work this part because the enumerations are not clear.

Reply:

It is actually the second major objective: "The second major objective of ESM-SnowMIP is to better quantify snow-related global climate feedbacks."

**Reviewer comment**:

P6 L4 - L10 & L20-L22 : these sentences fit better in Section 3 - Experimental Design, explaining why the gridded simulations have been augmented by single-point simulations. Also, suggest to add a sentence saying that there are basically two groups of snow models participating: those doing the gridded AND point experiments, and those only doing the latter. It is implicitly clear from Section 3 but would be good to make explicit.

Reply:

As suggested by the reviewer, these sentences were elevated to section 3: "Only climate models will be able to perform the global coupled simulations required for CMIP6, and their land-surface models will carry out global uncoupled simulations driven with a prescribed meteorological forcing. However, all models, that is, both those that are coupled to an atmospheric model and those that are not (i.e. standalone snow models), can perform the local uncoupled reference site simulations for ESM-SnowMIP at much lower computational expense. Models that have already completed the first round of reference site simulations, listed in Table 1, include land surface schemes (LSS) of CMIP6 models, standalone snow physics models, hydrological models and multi-physics ensemble models."

**Reviewer comment**:

P6 L18: please quantify number of sites and years.

Reply:

We replaced "the range of sites and the numbers of years simulated in ESM-SnowMIP far exceed those in similar experiments for SnowMIP and PILPS2d." with "the 136 site-years simulated in the first round of ESM-SnowMIP already far exceed similar experiments with 20 site-years in SnowMIP1, 19 site-years in SnowMIP2 and 18 years at one site in PILPS2d."

**Reviewer comment**:

P6 L20: Distinction between climate models and ESMs is superfluous? Nowadays, they basically mean the same thing, since the usual definition of ESM is a climate model with an active carbon cycle.

Reply:

We now write: "Only climate models will be able to perform the global coupled simulations required for CMIP6..."

**Reviewer comment**:

P6 L23: suggest to replace "sophisticated snow physics models" by "standalone snow models" since the word sophisticated is subjective.

Reply:

OK, done.

**Reviewer comment**:

P6 L26: at what frequency are these measurements typically available?

Reply:

This is extremely variable. We now write: "Measurements of snow water equivalent

(SWE) and depth, and thus also bulk snow density, are available for all of the reference sites at frequencies varying from hourly to monthly."

**Reviewer comment**:

P7 L1: This phrasing is not very formal. Do you mean to say that the onset of melt is both under- and overestimated by models?

Reply:

We reformulated this sentence. It now reads: "Some models have rather low albedos, leading to snow disappearing too early in the spring, and snow remains on the ground for too long in some other models."

**Reviewer comment**:

P7 L4: is this because there is no snow, thus no insulating effect?

Reply:

Yes. It is a time when snow-related errors from the previous cold season have no effects any more, while new snow is not on the ground yet. We clarify this by writing: "For both sites, there is a strong reduction in model temperature spread as soils cool in autumn, before the onset of snow cover and soil freezing."

**Reviewer comment**:

P7 L15-16: suggest to split into two sentences. Further, I would say that soil temperature is also dependent on the amount of meltwater refreezing and the refreezing depth (see e.g. Van Kampenhout et al., 2017)

Reply:

OK. We now write: "Snow mass balance is influenced by radiated, advected and conducted heat fluxes in the energy balance. Soil temperature is influenced by snow depth, thermal conductivity, amount of meltwater refreezing and refreezing depth (van Kampenhout et al. 2017)."

**Reviewer comment**:

P7 L28: Please add reference for the statement "which approximates the CMIP5 multimodel mean peak snow albedo" if you have one.

Reply:

We now write: "An experiment in which snow albedo is fixed to 0.7, which is a typical snow pre-melt albedo (Harding and Pomeroy 1996; Melloh et al. 2002; Wang et al. 2016), will enable evaluation of the effect of seasonal snow albedo variations and biases."

**Reviewer comment**:

P8, L11: Please explain briefly why the ensemble members react differently, for people that don't know FSM.

Reply:

We now write:

"FSM only has a single option for stability adjustment of the surface exchange coefficient, but feedbacks involving other processes make ensemble members respond differently to switching this option off, as seen in Figure 6."

**Reviewer comment**:

P8 L27-29: Unclear, please rewrite.

Reply:

We now write: "Without the insulating effect of snow, the soil freezes even in the relatively mild winters at Col de Porte. A cold bias persists into the spring and delays the melting of snow. There is a second trough in soil temperature differences between high thermal conductivity and reference simulations because energy is being used to

melt the late-lying snow rather than warming the soil."

**Reviewer comment**:

P8 L32: acronym GSWP3 is first used but not defined nor referenced

Reply:

It's now defined: "Meteorological variables in large-scale forcing datasets, such as the GSWP3 (Global Soil Water Project Phase 3) meteorological forcing data provided at 0.5° spatial resolution for LS3MIP (Kim et al., in preparation), would therefore be expected to be biased relative to in situ measurements at the sites even if they were perfect on the grid scale."

**Reviewer comment**:

P9 L2: suggest to avoid the words "overestimate" and "underestimated" because these have negative connotations. What you mean to say is that there is a differences between the grid cell means and the measurement site because they differ in elevation.

Reply:

OK. We now write: "Figure 8a shows that an FSM simulation for winter 2009-2010 at Col de Porte with GSWP3 driving data gives almost no snow accumulation; this is because temperature on the grid scale, because of the lower mean altitude, is higher than at Col de Porte, while total precipitation is lower and snowfall is much lower."

**Reviewer comment**:

P9 L3: is this coincidence, or is Sodankylä located in flat terrain? Probably good to mention.

Reply:

Sodankylä is in a really flat region. We now write: "Site and grid elevations for So-

Interactive
comment
dankylä, in contrast, only differ by 40 m because this site is situated in a flat area. The large-scale simulation shown in Figure 8b is not so strongly influenced by driving data biases."

**Reviewer comment**:

P9 L5 - L10: I would not call bias-correction a form of downscaling. A (statistical) downscaling procedure would use the climate data as is, then projecting that onto a high resolution topography using lapse rates and possibly repartitioning of precip

Reply:

OK. We now write "altitude-adjusted large-scale driving data" instead of "bias-adjusted large-scale driving data".

**Reviewer comment**:

P9 L17: SCF first used but not defined

Reply:

Now defined here: "Observation-based estimates of SWE and snow-cover fraction (SCF) are required..."

**Reviewer comment**:

P10 L3: unclear what is meant with "fully characterized bias and error"

Reply:

We clarified this. We now write: "A second reason to use a suite of analyses for MIP evaluation is that the bias and error of individual datasets vary with geographical location and datasets that perform well in some regions may perform more poorly in others. The lack of a clear "best" dataset provides minimal reason to favor one analysis over another. In fact, it has been explicitly demonstrated that combinations of products have both lower bias and RMSE than individual products when evaluated over domains

with in situ data (Schwaizer et al. 2016)."

**Reviewer comment**:

P10 L9 - L14: can we do even better by defining a unique threshold for each of the different snow products? Was this tested?

Reply:

This was tested as part of SnowPEx. Values differed somewhat depending on the domain and calender months over which each product was optimized but ranged between 3mm (e.g. GLDAS2) and 8mm (Crocus). There is also uncertainty in the values of the MEaSUREs data which was not considered in this process. While choosing a separate threshold for each dataset may reduce the spread in SCE among the component products, we are not convinced it would represent a real reduction in the uncertainty.

**Reviewer comment**:

P10 L16 - L22: Unclear; make more clear that LS3MIP is used as a baseline or control experiment and that ESM-SnowMIP adds sensitivity experiments on top of that. Reference Kim 2018 should be given earlier, when GSWP3 is introduced.

Reply:

OK. We now write: "The global land-only simulations planned in ESM-SnowMIP build on the reference historical land simulation (Land-Hist) currently carried out in the framework of the LS3MIP project.", and cite Kim (in preparation) earlier.

**Reviewer comment**:

P10 L25: and prescribed SCF? If not, mention that this is left up to the model (e.g. on P11 L4)

Reply:

We tried to clarify this sentence: "Here we propose a prescribed SWE experiment

to identify LSM biases that are linked to the parameterization of surface albedo as a function of the snow cover fraction, which in turn is very often a diagnostic function of the SWE which is prescribed in this experiment."

**Reviewer comment**:

L11 L2: "by less than 10% or so" : too vague

Reply:

OK. We now write "(by less than 10%)".

**Reviewer comment**:

P11, L9: on P9 L19 you wrote that "we have developed a blended dataset" which suggested that it was specifically developed for this MIP. This contrasts with the fact you now use the reference Mudryk 2015. Indicate if and how the blended dataset used is different from Mudryk 2015 and avoid inconsistencies.

Reply:

We have elected to use same five datasets described in Mudryk et al, 2015. Figures and captions have been updated accordingly. Text was clarified at line 23.

**Reviewer comment**:

P11 L10: section number missing.

Reply:

This was corrected (inserted "section 3.2.1").

**Reviewer comment**:

P11 L28: Aren't all ESM-SnowMIP gridded experiments?

Reply:

No, we do have site simulations, which are not gridded (section 3.1). Maybe we misunderstand the question.

**Reviewer comment**:

P12 L1 - L2: are these observational data for evaluation already known? If yes, mention them.

Reply:

We now write "in particular snow melt dates (to be obtained from the ensemble of large-scale SCE data)..."

**Reviewer comment**:

P12 L10: I don't see how the active layer depth can be diagnosed from the lowermost soil layer depth, which may vary across models.

Reply:

The sentence was misleading. The lowermost level I used to diagnose the permafrost extent. We now write: "In this global setting, simulated potential permafrost extent (that is, the permafrost extent in equilibrium with the prescribed climate and model setup; often also termed "near-surface permafrost", e.g. Lawrence et al. (2008)) will be diagnosed from the thermal state of the lowermost soil layer in the simulations."

**Reviewer comment**:

P12 L34: clarify what CCI 200 means

Reply:

Using the year 2000 will facilitate comparison with the GLC2000 land cover data here at ECCC. It also represents a rough mid-point over the period the dataset is available. We now write: "We thus suggest the use of the CCI land cover dataset for the year 2000 as the common land cover dataset from which to derive PFTs."

**Reviewer comment**:

P13, L9 - L18: Please clarify this part. In particular unclear is this sentence "This LS3MIP experiment uses (: : :) and prescribes these in the LFMIP-rmLC experiment." What is meant with "scenario simulation" (mind that scenario = future in CMIP terminology) and "context of a transient run".

Reply:

We tried to clarify this paragraph. It now reads: "ESM-SnowMIP proposes one coupled Tier 1 experiment, which serves the purpose of quantifying snow-related feedbacks in the global climate system on interannual time scales. It is designed to separate the effects of snow from the effects of snow and soil humidity, the combined effect being addressed by the LS3MIP Tier 1 coupled experiment LFMIP-rmLC. This LS3MIP experiment uses 30-year running mean land conditions (snow and soil humidity) as simulated in a reference transient climate change experiment, and prescribes these in a second simulation. In these runs, snow and soil moisture feedbacks on decadal and shorter timescales are muted. Comparing the LFMIP-rmLC simulation to the appropriate projection used for prescribing the land surface conditions allows identifying these feedbacks. In the context of a transient run, additional diagnoses of geographic shifts of land-atmosphere coupling hotspots (Seneviratne et al. 2006) and changes in potential predictability related to land surface (Dirmeyer et al. 2013) can be carried out. The SnowMIP-rmLC will isolate the effects of snow-atmosphere coupling by prescribing only the soil humidity state from the reference simulation, not the entire surface state as in LFMIP-rmLC."

**Reviewer comment**:

P13 L19: Unclear whether the run being proposed here is SnowMIP-rmLC? Make explicit.

Reply:

[Figure]

The paragraph was rewritten (see above) to clarify this.

**Reviewer comment**:

P14: this page breaks the general style of the manuscript by using bullet points extensively.

Reply:

The style was harmonized.

**Reviewer comment**:

P14 L27 - L32: suggest to move this to the end of the section, P15 L3.

Reply:

Done.

**Reviewer comment**:

P16, L16: it should be made more clear whether the possible future extensions are intended for a follow-up project (ESM-SnowMIP Phase 2 or whatever) or as an integral part of the current effort.

Reply:

We changed the name of the section (now "Possible actions for future follow-up projects") and slightly changed the wording to make this point clear.

**Reviewer comment**:

P16, L24: and bottom heat flux, presence of salt

Reply:

OK. We now write: "The physical properties of snow on sea ice are linked to low accumulation rates and strong vertical temperature gradients due to bottom heat flux,

its spatial heterogeneity, its peculiar evolution in summer leading to melt ponds on sea ice due to inhibited drainage of meltwater, and the presence of salt."

**Reviewer comment**:

P18 Many citations are not up to date and contain either the word "Received" (e.g. L7) or they point to a discussions paper (e.g. L22). Next to that the use of URLs is not consistent, e.g. L25 contains an URL that basically repeats the DOI. Suggest to remove all URLs and stick to DOI. Whole reference list needs a cleanup like this.

Reply:

We cleaned up the reference list.

**Reviewer comment**:

P21, L20: this work is not available under any DOI so reference should be removed.

Reply:

OK.

**Reviewer comment**:

P25, Figure 1: over what period was the average computed?

Reply:

This depends on the records length that varies from site to site. We modified the caption of Figure 1 accordingly:

"Figure 1: Winter (DJF) temperatures and annual snowfall averaged over the forcing data periods at the ESM-SnowMIP reference sites (see Table 3)."

**Reviewer comment**:

P27 Figure 5 - 7: suggest to replace "ensembles" with "ensemble members" and mention in each case how many ensemble members are present in the graph.

[Figure]

Reply:

We changed the Figure captions 5-7 accordingly and provided the required information:

"Figure 5: Differences between the 32 FSM ensemble members in fixed albedo and reference simulations for Col de Porte, averaged over October 1994 to September 2014.

Figure 6: As Figure 5, but for differences between 16 FSM ensemble members with fixed surface exchange coefficients and 16 with variable coefficients.

Figure 7: As Figure 5, but for differences between the 32 FSM ensemble members in simulations without thermal insulation by snow and reference simulations."

**Reviewer comment**:

P27, Figure 5: What is meant with "Correct prescription"? Further, the word "period" is missing.

Reply:

We meant the reference simulation. This is clarified in the revised version.

**Reviewer comment**:

P29, Figure 10: The 7th dataset, MERRA, appears to be missing from the graph

Reply:

The figure was redrawn.

**Reviewer comment**:

P31, Table 2: suggest to put site abbreviations / tags in separate column for quick reference.

Reply:

The table was modified accordingly.

The remaining "Technical corrections" were all very minor suggestions (typos etc.) that were implemented. Again, we thank the reviewer for his detailed work.